# Rapid glacier retreat and downwasting throughout the European Alps in the early 21st century

Christian Sommer [1✉], Philipp Malz[1], Thorsten C. Seehaus [1], Stefan Lippl [1], Michael Zemp [2] & Matthias H. Braun [1]

Mountain glaciers are known to be strongly affected by global climate change. Here we compute temporally consistent changes in glacier area, surface elevation and ice mass over the entire European Alps between 2000 and 2014. We apply remote sensing techniques on an extensive database of optical and radar imagery covering 93% of the total Alpine glacier volume. Our results reveal rapid glacier retreat across the Alps ($-39\,\mathrm{km^2\,a^{-1}}$) with regionally variable ice thickness changes ($-0.5$ to $-0.9\,\mathrm{m\,a^{-1}}$). The strongest downwasting is observed in the Swiss Glarus and Lepontine Alps with specific mass change rates up to $-1.03\,\mathrm{m.w.e.\,a^{-1}}$. For the entire Alps a mass loss of $1.3 \pm 0.2\,\mathrm{Gt\,a^{-1}}$ (2000–2014) is estimated. Compared to previous studies, our estimated mass changes are similar for the central Alps, but less negative for the lower mountain ranges. These observations provide important information for future research on various socio-economic impacts like water resource management, risk assessments and tourism.

[1] Institut für Geographie, Friedrich-Alexander-Universität Erlangen-Nürnberg, Wetterkreuz 15, 91058 Erlangen, Germany. [2] Department of Geography, University of Zurich, Winterthurerstr. 190, 8057 Zurich, Switzerland. ✉email: chris.sommer@fau.de

Substantial retreat and downwasting of mountain glaciers due to global warming have been observed worldwide[1]. In the European Alps, glaciers have been retreating since the Little Ice Age (~1850)[2–4] and future ice volumes are predicted to be largely reduced[5–7]. During previous decades, accelerated glacier shrinkage has been reported[8–10]. Mass-change rates were close to $-1$ m.w.e. a$^{-1}$ during the first 5 years of the 21st century[11]. The ongoing reduction of glacier volume raises challenges for water supply during dry periods, civil security, and tourism[12].

Mountain regions are frequently described as "water towers"[13]. Seasonal shifts in glacier meltwater discharge can have widespread impact on runoff during dry periods[14–16]. In the Alps, meltwater contributes to late-summer runoff when seasonal snow cover is minimal[17]. During 1908–2008 glacier discharge contributed ~20% to August runoff of the Rhone and Po rivers[17]. However, maximum runoff from glacier long-term storage ("peak water") has already been or will be reached in the coming decades[14].

On a regional scale, changes in seasonal runoff affect the production of renewable energy in Alpine countries and require adaptive strategies for hydropower[18]. As another important economy, summer tourism partly relies on the scenery of the glacierized Alpine landscape. Shrinking glaciers affect tourism by changing the shape of the landscape and frequency of natural hazards[19].

In order to predict future water availability, information on past glacier changes are essential to improve simulations on glacier evolution and thereby also future runoff projections[16]. However, most glacier change studies in the Alps focus on large-scale catchments or a limited number of in situ measurements which might lack representativeness. A comprehensive and methodologically consistent cross-border analysis on glacier changes over the entire Alps in the early 21st century with the same observation interval is so far missing.

Here, we provide glacier-specific area and elevation measurements and regional mass changes for all Alpine regions. We compare digital elevation models (DEM) from two radar interferometry missions and temporal consistent optical satellite images on glacier area changes. We apply this approach for the intervals 2000–2012 and 2000–2014. Our results represent a detailed assessment of glacier-specific mass changes throughout the entire Alps.

## Results

**Glacier change measurement approach.** We use synthetic aperture radar (SAR) data from the Shuttle Radar Topography Mission (SRTM) in 2000 and the TerraSAR-X-Add-on for Digital Elevation measurements mission (TanDEM-X) as well as optical imagery of the Landsat program to measure glacier changes. We compare specifically generated TanDEM-X DEMs (~270 DEMs) from two acquisition periods (2011–2012, 2013–2014) covering the entire Alps and the SRTM DEM to analyse spatial variations of elevation changes. A crucial advantage of our approach is the combination of temporally consistent area and elevation measurements, which improve the accuracy of mass change estimates. Processing of elevation changes was adopted from previous studies[20–23]. A strength of the interferometric SAR compared with optical sensors is that radar acquisitions are not influenced by clouds or oversaturation by highly reflective surfaces (e.g., glacier accumulation areas). To compensate for potential SAR signal penetration into the winter glacier surface, we apply an additional altitude-dependent correction. For the area assessment we compute optical band ratios[22,24,25] from Landsat imagery (185 scenes, 1999–2001, 2011, 2013–2015). Our glacier change measurements have been aggregated over different regional subdivisions. The

Western and Eastern Alps, divided by the Rhine Valley and Splügen Pass, include all glaciers whereas regions 01–10 represent smaller subdivisions according to the International Standardized Mountain Subdivision of the Alps (IMSA)[26] with at least 10 km² glacierized area (Fig. 1c).

**Alpine-wide glacier shrinkage and downwasting.** Highly negative mean elevation changes ($< -0.6$ m a$^{-1}$, Fig. 1a) are recorded for both larger subregions in the Western (regions 01–06) and Eastern Alps (regions 07–10) during 2000–2014 (Fig. 2 and Table 1). Balanced conditions (elevation change ~0 m a$^{-1}$) are observed above ~3500 m a.s.l. for the Graian, Pennine, and Bernese Alps with no region having significantly positive values even at highest glacier elevations. In many regions, change rates are negative throughout all altitudes, indicating the loss of former accumulation areas and thinning over the entire glacier. Particularly, areas below ~2000 m a.s.l. experience average regional surface lowering of up to 5 m a$^{-1}$ in the Graian, Bernese, and Glarus Alps. Glacier-specific change rates can be even more negative (e.g., $< -8$ m a$^{-1}$ at terminus of Grosser Aletsch, Bernese Alps) caused by the complete downwasting of frontal areas during the observation period. Area (Fig. 1b) and mass change (Fig. 1c) are controlled by the regionally different extents of glaciers. The highest absolute area reductions are found in the Bernese, Pennine, and Graian Alps which include the largest glacier areas of the Alps. The overall retreat is ~$39 \pm 9$ km² a$^{-1}$, corresponding to an area loss rate of ~1.8% a$^{-1}$ between 2000 and 2014 (for regional changes see Supplementary Table 1, Supplementary Figs. 1 & 2). Highest mass loss rates ($< -0.2$ Gt a$^{-1}$) are measured in the Western Alps (Bernese, Pennine Alps).

The highest mass losses are measured in the Glarus, Lepontine, and Rhaetian Alps during 2000–2012 and 2000–2014. The Glarus Alps show the lowest specific mass-change rate (~$-1$ m.w.e. a$^{-1}$), followed by the Lepontine Alps. The Western and Southern Rhaetian Alps are slightly less negative ($< -0.8$ m.w.e. a$^{-1}$). The largest subregions in the Western Alps show more heterogeneous patterns. While the Bernese Alps reveal a strongly negative mass-change rate ($< -0.8$ m.w.e. a$^{-1}$), less pronounced glacier wastage ($> -0.7$ m.w.e. a$^{-1}$) is recorded in the Graian and Pennine Alps. The higher losses in the Bernese Alps are probably driven by the altitudinal distribution of the glacierized areas. The Graian and Pennine Alps have considerable glacier areas above 3500 m a.s.l. while large areas of the Bernese Alps are located at lower altitudes. In the most Western and Eastern subregions (Dauphiné and Tauern Alps) slightly less negative mass-change rates are measured.

Comparing both periods, specific mass change is similar for 2000–2012 ($-0.71 \pm 0.14$ m.w.e. a$^{-1}$) and 2000–2014 ($-0.70 \pm 0.13$ m.w.e. a$^{-1}$) with slightly more negative values in the Graian, Bernese, Lepontine, and Tauern Alps during the former period. In the remaining regions, differences between 2000–2012 and 2000–2014 are within the uncertainty ranges ($<0.05$ m.w.e. a$^{-1}$, Methods section) and might be partially related to the regional TanDEM-X acquisition dates and prevailing climatic conditions. On a country-level, similar specific mass changes are observed in the French, Swiss, and Austrian Alps ($< -0.7$ m.w.e. a$^{-1}$) while mass loss in the Italian Alps is slightly less negative (~$-0.6$ m.w.e. a$^{-1}$, Table 1).

For 2000–2014 we assessed the differences in specific mass change when changes in glacier area during the observation period are neglected. Assuming a constant area (based on Randolph Glacier Inventory V6.0[27]), the total mass-change rate would be underestimated by ~14%.

Average elevation changes of Western and Eastern Alps appear to be linked to altitude and glacier size with strongly negative

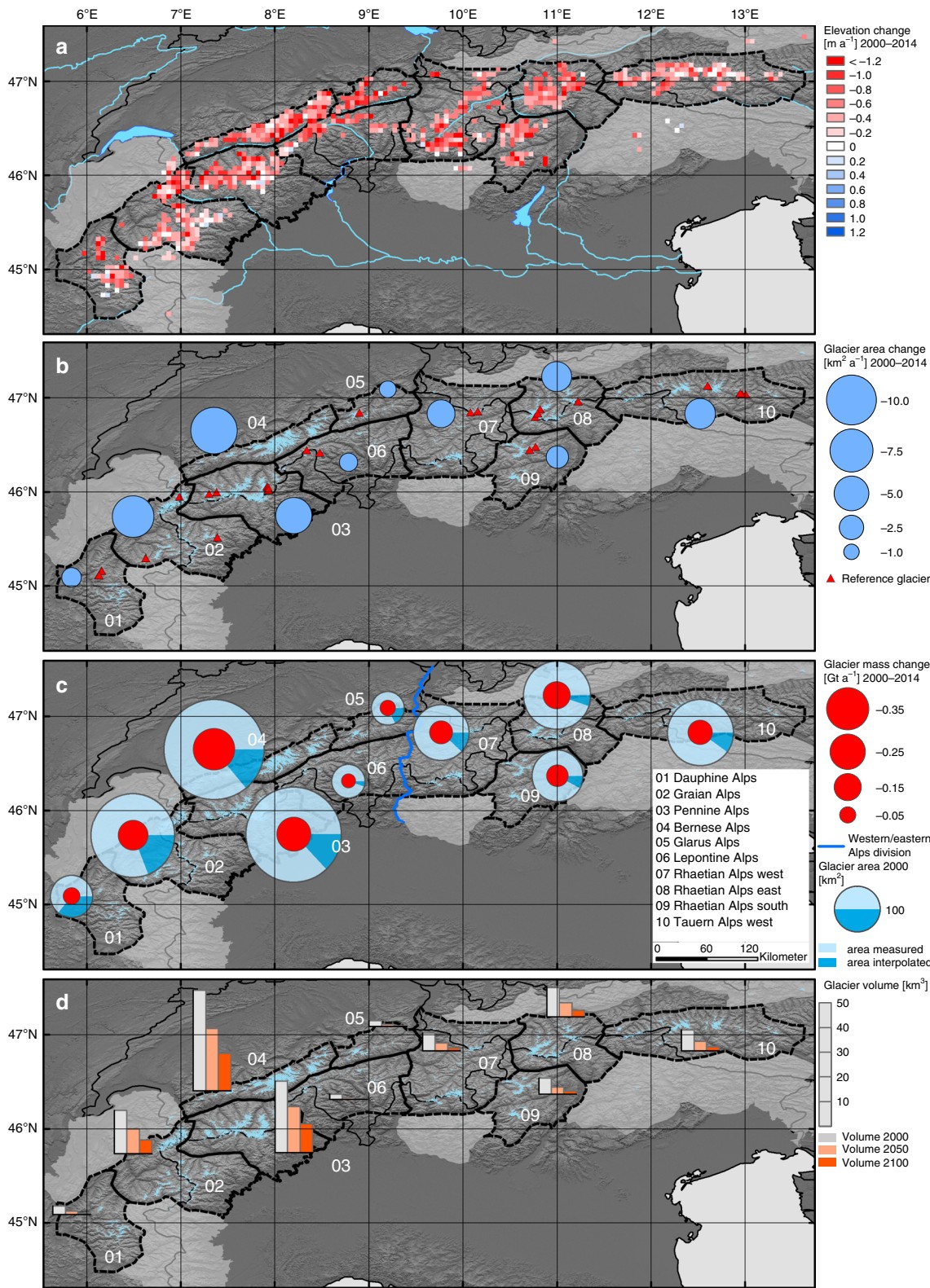

**Fig. 1 Glacier area, elevation, and mass change of the European Alps 2000–2014.** Black dotted outlines indicate regional subdivisions (according to IMSA mountain range classification[26]). **a** Average glacier elevation change rates within 0.05° grid cells, cells with <0.05 km² glacier area are not displayed. **b** Glacier area change rates of each subregion. Change values of regions with <10 km² of glacier area are not shown (lightgray areas). Red triangles are glaciers with continuous glaciological mass balance measurements 2000–2014 shown in Fig. 3. **c** Total glacier areas of each subregion and respective fractions of measured and interpolated areas (corresponding to DEM coverage) indicated as blue pie charts. Respective mass-change rates are plotted as red dots. **d** Early 21st century ice volume[28] (gray bars) and remaining ice volumes in 2050 (lightorange bars) and 2100 (darkorange bars) based on relative volume change rates between 2000 and 2014 (Background: SRTM hillshade).

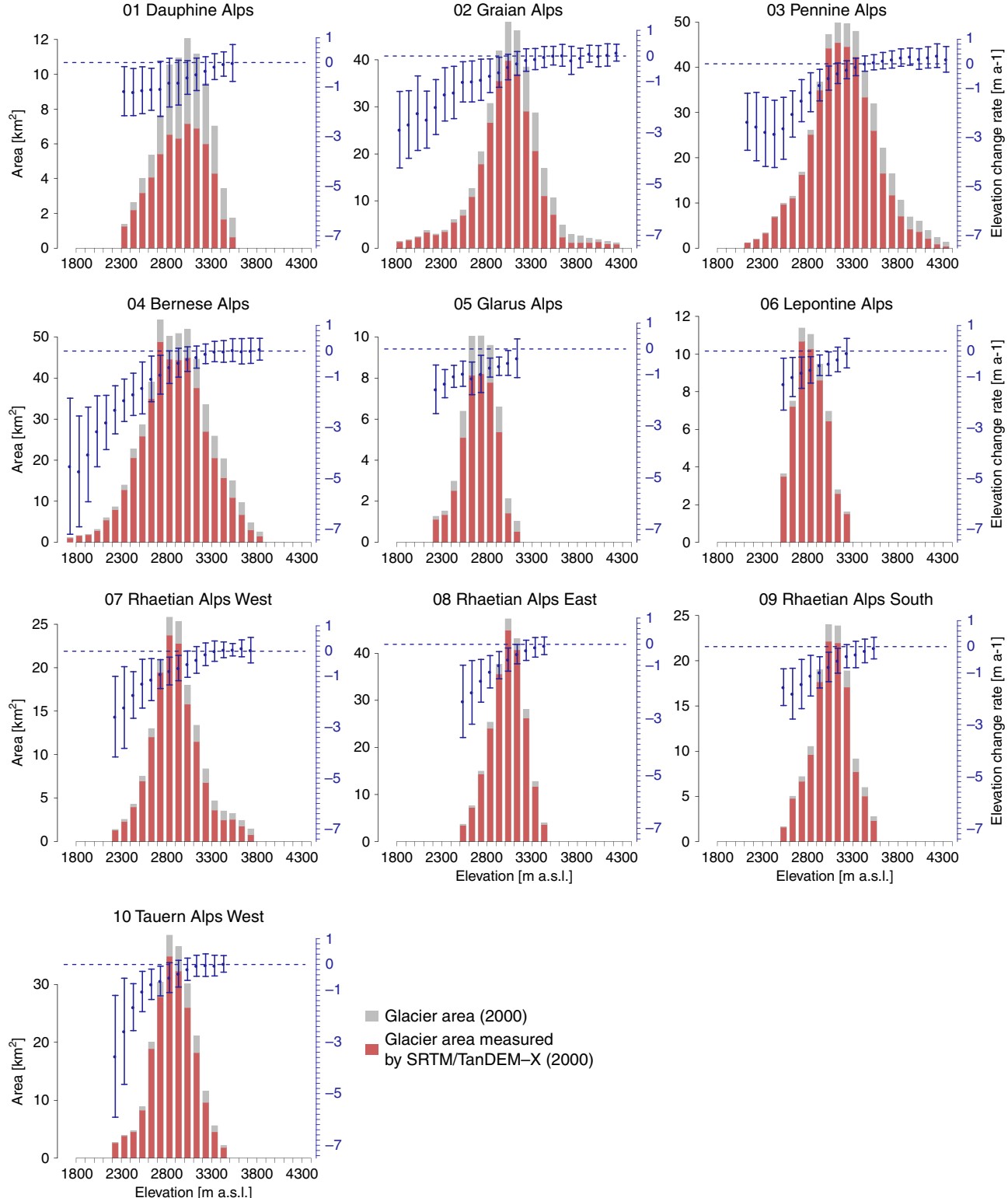

**Fig. 2 Altitudinal distribution of glacier surface elevation change rates 2000–2014 and area per region.** The total glacierized area (in 2000) within each 100 m elevation bin is indicated by gray bars and the respective fraction of area with elevation change measurements (TanDEM-X & SRTM) by red bars. Blue dots and error bars show the measured average elevation change and respective normalized median absolute deviation per elevation bin. Note: different scaling of the glacier area axes.

values at lower altitudes and smaller changes above 3500 m a.s.l. (Supplementary Fig. 3). Glaciers in the Western Alps show a large range of average change rates, varying between −4 and 0.5 m a⁻¹.

We attribute this to the large valley glaciers ranging from ~1400 to 4700 m a.s.l. while in the Eastern Alps accumulation zones are located at lower altitudes and glacier termini reach less far down

**Table 1 Regional area, elevation, mass (Gt a⁻¹) and specific mass change (m.w.e. a⁻¹) of glaciers in the European Alps for two observation periods (2000–2012, 2000–2014).**

| ID | Region name | Glacier area (km²) 2000 | Area measured mean (%) | Mean elevation change rate (m a⁻¹) 2000–2012 | Mean elevation change rate (m a⁻¹) 2000–2014 | Mass-change rate (Gt a⁻¹) 2000–2012 | Mass-change rate (Gt a⁻¹) 2000–2014 | Mass-change rate (m.w.e. a⁻¹) 2000–2012 | Mass-change rate (m.w.e. a⁻¹) 2000–2014 | RGI6.0-based mass-change rate (m.w.e. a⁻¹) 2000–2014 | Median date of TanDEM-X 2012 | Median date of TanDEM-X 2014 |
|---|---|---|---|---|---|---|---|---|---|---|---|---|
| 01 | Dauphiné Alps | 86.8 | 63.5 | −0.562 ± 0.225 | −0.620 ± 0.196 | −0.054 ± 0.021 | −0.056 ± 0.018 | −0.690 ± 0.266 | −0.739 ± 0.237 | −0.597 ± 0.192 | 2012-01-24 | 2013-12-26 |
| 02 | Graian Alps | 340.8 | 80.7 | −0.648 ± 0.096 | −0.520 ± 0.070 | −0.220 ± 0.042 | −0.176 ± 0.033 | −0.723 ± 0.141 | −0.604 ± 0.116 | −0.519 ± 0.096 | 2012-02-25 | 2014-04-14 |
| 03 | Pennine Alps | 442.1 | 86.7 | −0.460 ± 0.060 | −0.493 ± 0.054 | −0.224 ± 0.053 | −0.227 ± 0.046 | −0.544 ± 0.128 | −0.562 ± 0.114 | −0.496 ± 0.096 | 2012-03-08 | 2014-02-27 |
| 04 | Bernese Alps | 485.7 | 85.9 | −0.736 ± 0.068 | −0.719 ± 0.059 | −0.362 ± 0.064 | −0.345 ± 0.056 | −0.828 ± 0.148 | −0.806 ± 0.131 | −0.703 ± 0.107 | 2012-02-20 | 2014-03-20 |
| 05 | Glarus Alps | 53.7 | 82.3 | −0.859 ± 0.155 | −0.901 ± 0.154 | −0.045 ± 0.010 | −0.046 ± 0.009 | −0.985 ± 0.211 | −1.031 ± 0.208 | −0.846 ± 0.145 | 2011-12-16 | 2013-12-20 |
| 06 | Lepontine Alps | 54.6 | 95.0 | −0.795 ± 0.160 | −0.732 ± 0.140 | −0.045 ± 0.011 | −0.040 ± 0.009 | −0.974 ± 0.233 | −0.889 ± 0.204 | −0.731 ± 0.155 | 2012-02-20 | 2013-11-30 |
| 07 | Rhaetian Alps West | 149.6 | 87.7 | −0.705 ± 0.124 | −0.710 ± 0.110 | −0.106 ± 0.023 | −0.106 ± 0.021 | −0.809 ± 0.176 | −0.841 ± 0.167 | −0.708 ± 0.121 | 2012-01-24 | 2014-01-18 |
| 08 | Rhaetian Alps East | 220.4 | 95.0 | −0.638 ± 0.072 | −0.671 ± 0.065 | −0.148 ± 0.029 | −0.149 ± 0.026 | −0.739 ± 0.147 | −0.770 ± 0.134 | −0.662 ± 0.111 | 2011-12-23 | 2013-12-26 |
| 09 | Rhaetian Alps South | 126.6 | 91.8 | −0.749 ± 0.081 | −0.725 ± 0.072 | −0.096 ± 0.017 | −0.091 ± 0.015 | −0.831 ± 0.152 | −0.814 ± 0.138 | −0.723 ± 0.111 | 2012-03-20 | 2014-03-26 |
| 10 | Tauern Alps | 211.2 | 90.1 | −0.545 ± 0.132 | −0.526 ± 0.120 | −0.124 ± 0.035 | −0.117 ± 0.031 | −0.660 ± 0.185 | −0.635 ± 0.168 | −0.557 ± 0.132 | 2012-02-21 | 2014-03-21 |
| A | French Alps | 245.9 | 76.1 | −0.705 ± 0.141 | −0.650 ± 0.109 | −0.177 ± 0.041 | −0.159 ± 0.033 | −0.820 ± 0.193 | −0.767 ± 0.162 | −0.636 ± 0.130 | 2012-01-24 | 2014-04-14 |
| B | Italian Alps | 455.1 | 84.4 | −0.567 ± 0.110 | −0.528 ± 0.100 | −0.263 ± 0.058 | −0.240 ± 0.052 | −0.628 ± 0.141 | −0.596 ± 0.131 | −0.524 ± 0.104 | 2012-02-25 | 2014-02-27 |
| C | Swiss Alps | 1087.8 | 86.7 | −0.663 ± 0.083 | −0.659 ± 0.070 | −0.727 ± 0.137 | −0.706 ± 0.119 | −0.741 ± 0.141 | −0.732 ± 0.125 | −0.643 ± 0.100 | 2012-03-20 | 2014-02-27 |
| D | Austrian Alps | 403.9 | 92.7 | −0.597 ± 0.111 | −0.609 ± 0.104 | −0.254 ± 0.061 | −0.250 ± 0.054 | −0.709 ± 0.171 | −0.718 ± 0.157 | −0.629 ± 0.127 | 2011-12-23 | 2013-12-26 |
| I | Western Alps | 1474.4 | 83.6 | −0.627 ± 0.087 | −0.606 ± 0.071 | −0.931 ± 0.178 | −0.877 ± 0.151 | −0.700 ± 0.135 | −0.676 ± 0.118 | −0.582 ± 0.094 | 2012-02-20 | 2014-02-27 |
| II | Eastern Alps | 719.0 | 91.2 | −0.646 ± 0.107 | −0.648 ± 0.099 | −0.474 ± 0.101 | −0.465 ± 0.091 | −0.737 ± 0.158 | −0.744 ± 0.147 | −0.647 ± 0.115 | 2012-01-13 | 2013-12-26 |
| | Alps total | 2193.4 | 86.1 | −0.633 ± 0.094 | −0.620 ± 0.080 | −1.405 ± 0.279 | −1.342 ± 0.242 | −0.712 ± 0.143 | −0.698 ± 0.128 | −0.603 ± 0.101 | 2012-02-08 | 2014-02-05 |

Glacier volumes are converted to mass with an average density of 850 kg m⁻³. Area measured mean is the average regional coverage of glacier areas in percent by TanDEM-X and SRTM during 2000–2012 and 2000–2014. Specific mass changes based on glacier areas from Randolph Glacier Inventory V6.0[27] are also provided. TanDEM-X dates refer to the regional median date of all TanDEM-X measurements covering glacierized areas.

valley. Area changes show a similar pattern as surface elevation changes with the largest reductions at ~3000 m a.s.l. However, retreating areas are observed at most altitudes.

We demonstrate the vulnerability to an imminent glacier vanishing by imposing the present mass-change rates (2000–2014) on total estimated ice volumes[28] for each region. Figure 1d shows the glacier volume (~130 km³) at beginning and the proportion of ice that would survive under current regional mass-change rates ($−1.0$ to $−2.3\%\,a^{-1}$) over the course of the 21st century. This approach does not consider dynamic adjustments nor climate projections or any other factors which can only be achieved by respective modeling attempts[6]. Nevertheless, our extrapolated values match well with model projections[6] and show that the lower Alpine mountain ranges would be almost ice-free by the end of this century. The remaining glacier volume of the entire Alps would be approximately one-third compared with the volume at beginning of the 21st century. Larger amounts of ice (>10 km³) would only be left in the Pennine and Bernese Alps whereas the Dauphiné, Glarus, and Lepontine Alps are prone to be nearly ice-free within this century.

**Comparison with glaciological measurements.** The Alps have one of the densest in situ measurement networks worldwide with several glaciers measured for more than 30 years[29]. The glaciological records benefit from a high temporal resolution but can be biased by the small number of point measurements and issues related to the extrapolation to glacier-wide balances. Therefore, recent studies have adjusted glaciological mass balances with geodetic measurements, resulting in less negative values[1,30]. Moreover, in situ measurements are often limited to a small number of accessible glaciers and hence might not be representative for large regions.

Figure 3 shows the average annual glaciological mass balances of 25 glaciers with continuous measurements in comparison with the respective local and regional geodetic values. For most glaciers, the glaciological mass balance is similar or slightly more negative during 1999/00–2010/11 than the geodetic mass change. Overall, the geodetic measurements derived from SRTM and TanDEM-X are less negative than the glaciological ones (mean difference ~0.18 m.w.e. $a^{-1}$) particularly due to five glaciers (Ciardoney, Saint Sorlin, Fontana Bianca, Argentière, and Wurtenkees) which differ substantially. Regarding the geodetic estimates, those discrepancies might be related to the very small extents of some glaciers (Ciardoney, Fontana Bianca, Wurtenkees <1 km²) and biases in the radar acquisitions (e.g., shadow effects) or local climatic conditions (snow accumulation) at the acquisition time. However, there are also geodetic measurements which are within the uncertainty range of the glaciological measurements or even more negative.

For the Pennine, Glarus, Lepontine, Western and Eastern Rhaetian, and Tauern Alps, the glaciological measurements are close to the regional geodetic value. Systematic differences are found in the Dauphiné, Graian, and Southern Rhaetian Alps where the glaciological measurements are consistently more negative than the geodetic average. Nevertheless, the regional geodetic mass-change rates appear to be close to the area-weighted glaciological average in regions with a larger number of measurements (Pennine, Eastern Rhaetian, and Tauern Alps). There are, however, also glaciers which show more positive or negative glaciological values than the regional average (Supplementary Table 2) and might not be representative for the respective region without adjustment[1,30]. Our spatially comprehensive measurements provide a database for such an extended calibration of glaciological records throughout the Alps.

**Comparison with other geodetic observations**. Previous geodetic estimates showed strongly negative glacier mass changes in Switzerland (Supplementary Fig. 4), Austria, France, and Italy. For the period 1985–1999, the mass change of 786 Swiss glaciers was calculated from DEM differencing[31]. Proposing two approaches, cumulative mass changes of −7.0 m.w.e. (∼ −0.5 m.w.e. a⁻¹) and −10.95 m.w.e. (∼ −0.78 m.w.e. a⁻¹) were reported (with conversion factor 900 kg m⁻³). The first value is less negative while the second is similar to the rate of 2000–2012 for all Swiss glaciers (−0.74 ± 0.14 m.w.e. a⁻¹, with 850 kg m⁻³) of this study. For all Swiss glaciers, a change rate of −0.62 ± 0.07 m.w.e. a⁻¹ (1980–2010) was found[9]. We compute an even more negative rate for the Swiss Alps (−0.74 ± 0.14 m.w.e. a⁻¹) which is probably linked to warmer summer temperatures in the 21st century. However, local climatic drivers and interpretation of variability are more complex[32]. For the Grosser Aletsch glacier a detailed comparison of ICESat altimetry and different DEMs revealed a rate of −0.92 ± 0.18 m.w.e. a⁻¹ (2003–2009)[33]. This is very close to our estimate of −1.04 ± 0.16 m.w.e. a⁻¹ (2000–2012). In the Austrian Ötztal Alps, an ice volume loss of 1.0 km³ (∼ −0.11 km³ a⁻¹) between 1997 and 2006 was reported[10]. Our TanDEM-X–SRTM results show a very similar rate of −0.10 ± 0.02 km³ a⁻¹ for 2000–2012 (subset of Austrian Alps). Using high- and medium-resolution optical DEMs, a highly negative geodetic mass-change rate of −1.04 ± 0.23 m.w.e. a⁻¹ was found for the Mont Blanc massif between 2003 and 2012[34]. Our measurements for this subset of the Graian Alps are less negative (2000–2012: −0.78 ± 0.14 m.w.e. a⁻¹) despite the overlapping period. This region is characterized by high altitudes and rugged topography with a relatively low DEM coverage (<70%) by SRTM and TanDEM-X which might explain the differences. In the Italian Ortles-Cevedale group, a mass loss rate of 0.69 ± 0.12 m.w.e. a⁻¹ (1981–2007) was found[35]. Our measurement for 2000–2012 of the identical glaciers

(subset of region 09) is slightly lower (−0.73 ± 0.16 m.w.e. a⁻¹), probably due to more negative mass change in recent years.

Few mass change estimates are available throughout the entire Alps (Supplementary Fig. 5). Average specific mass changes of −0.31 ± 0.04 m.w.e. a⁻¹ (1900–2011) and −0.99 m.w.e. a⁻¹ (2000–2010) were estimated[8], based on records at 50 Swiss glaciers and extrapolation. Our measurements for 2000–2012 are 30% less negative than the extrapolated value for 2000–2010. We attribute the deviation to the differences in mass change between the high-altitude ranges and surrounding lower regions. Most previous observations were derived from the largely glacierized mountain ranges of France, Switzerland, and Austria. Our regional measurements show similar values in those regions, while mass changes in the lower Western and Eastern Alps are smaller.

For all glaciers in the Alps and Pyrenees, a recent study[1] estimated a mass change of −0.87 ± 0.07 m.w.e. a⁻¹ (2006–2016), using extrapolation based on glaciological and geodetic samples. Our results for the Alps are slightly less negative, most likely due to the different measured areas and periods. In particular, the years after 2010 had higher average summer temperatures than previous years (Supplementary Fig. 6), resulting in more negative mass change of many glaciers (Supplementary Fig. 7).

Our regional estimates of the period 2000–2014 represent the first Alpine-wide glacier mass change assessment and reveal widespread surface thinning even in the most upper reaches of the lower Alpine mountain ranges. The total mass loss is 1.3 ± 0.2 Gt a⁻¹ since 2000, corresponding to approximately −1.2% a⁻¹ of the glacier volume at beginning of the 21st century. The strongest contributors are the ablation zones of large valley glaciers of the high-mountain Swiss and Austrian Alps which are most out of balance and still adapting to present-day climate[36]. However, the actual amount of local surface lowering and ice melt

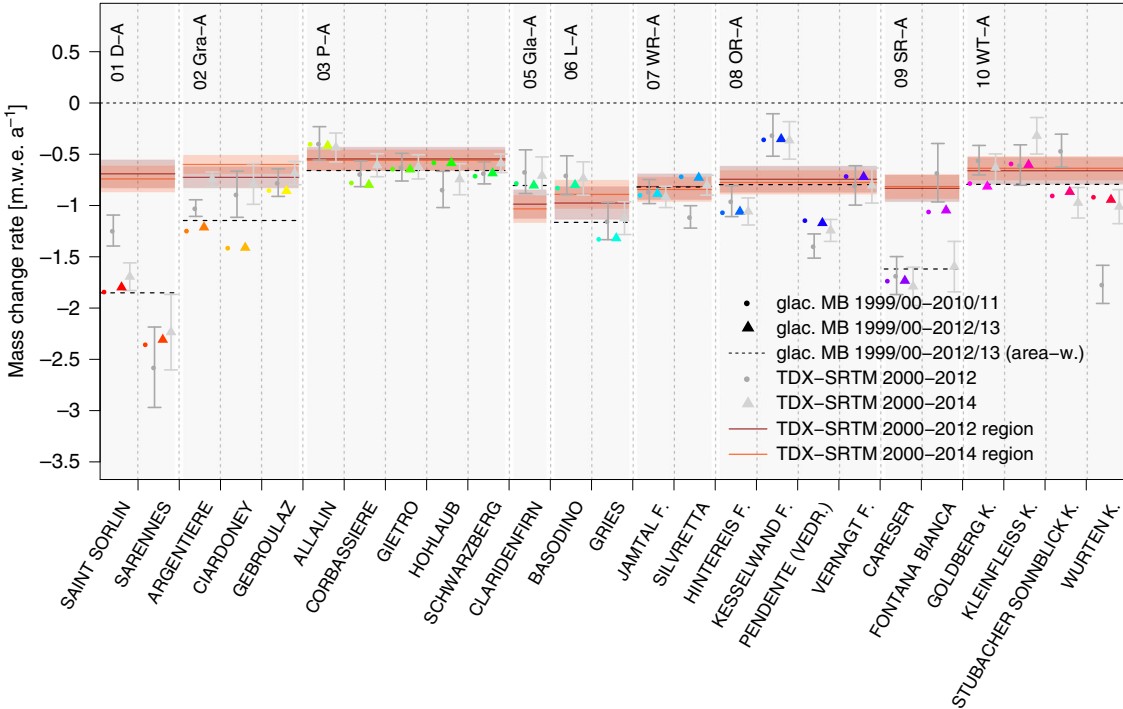

**Fig. 3 Average annual mass-change rates of glaciers[29] with continuous glaciological mass-change measurements between 1999/00–2010/11 (dots) and 1999/00–2012/13 (triangles), grouped accordingly to their geographic location and the regional subdivisions of this study.** Black dotted lines indicate the area-weighted regional values from glaciological measurements (1999/00–2012/13). Respective geodetic mass-change measurements of each glacier are shown as gray dots (2000–2012) and gray triangles (2000–2014). Red lines and error boxes indicate the average mass-change rate from TanDEM-X–SRTM of all glaciers of the associated subregion for the intervals 2000–2012 and 2000–2014.

at the frontal areas is probably even higher as major glacier retreat and complete deglacierisation was observed during the study period. The results of this study can be used to calibrate and validate mass-change models of glaciers in the Alps and improve hydrological projections.

## Methods

**Regional subdivisions**. We use the International Standardized Mountain Subdivision of the Alps (ISMSA)[26] classification to define glacier regions in the European Alps. The ISMSA combines historical mountain range subdivisions of the adjacent countries and the division into Western and Eastern Alps, roughly divided by the Alpine Rhine Valley, Splügen Pass, and Lake Como. We calculate geodetic glacier mass-change rates for the two large divisions Western and Eastern Alps, and ten smaller glacierized subregions (see Fig. 1), excluding regions with very small glacierized areas (<10 km²).

The French Dauphiné Alps (01 D-A) encompass the most western glacierized regions of the European Alps. The majority of glaciers are small (<0.5 km²) and located close to the Massif des Écrins. The Graian Alps (02 Gra-A) include glaciers in France, Italy, and Switzerland (>340 km²). The largest glaciers are found at the Mont Blanc massif, surrounding the highest peak of the Alps (Mont Blanc, 4808 m). The Pennine Alps (03 P-A) comprise the southern part of the canton of Valais (Switzerland) and the Aosta Valley (Italy). The second highest peak of the Alps (Dufourspitze, 4634 m) is located in this region and also the second largest glacier coverage (>440 km²). The Bernese Alps (04 B-A) are part of the canton of Bern and canton of Valais, Switzerland. The largest glacier of the Alps (Grosser Aletsch, 84 km² in 2000) is located within the Jungfrau-Aletsch mountain range with several summits above 4000 m. The Bernese Alps also comprise the largest glacierized area in this study (>480 km²). The Glarus (05 Gla-A) and Lepontine Alps (06 L-A) border the Bernese Alps and Pennine Alps to the East, respectively. The glacierized areas are significantly smaller than in the adjacent regions (<60 km²). The border between eastern Glarus and Lepontine Alps and Rhaetian Alps marks also the transition from Western to Eastern Alps. The Rhaetian Alps stretch across parts of Switzerland, Austria, and Italy. The majority of glacierized areas are located in the Bernina Range between Switzerland and Northern Italy (07 WR-A), Ötztal, Austria (08 OR-A), and Stelvio National Park, Northern Italy (09 SR-A). The Tauern Alps (10 WT-A) are the most eastern glacier region and include the largest glacier of the Eastern Alps (Pasterze, 18 km² in 2000) and several other medium-sized glaciers surrounding the peaks of Großvenediger (3657 m) and Großglockner (3798 m). Average elevations and elevation ranges of glaciers in each subregion are summarized in Supplementary Table 1.

**Area change calculation**. Glacier outlines are calculated for three dates (2000, 2011, 2014) contemporanous with available DEM datasets. To avoid seasonal snow and extensive cloud coverage, late-summer (Aug–Sep) Landsat images were selected, which represent the minimum glacier area at the end of the ablation period in the Alps. We use a total of 185 scenes from 1999–2001 (L5 TM & L7 ETM+), 2011 (L5 TM), and 2013–2015 (L8 OLI) to compute respective glacier areas corresponding to our DEMs. Band ratios (red/shortwave-infrared)[24] are created for each scene and converted to binary raster masks by applying manually selected thresholds. Thereafter, each raster is vectorized to create glacier polygons. To preserve comparability, ice divides between individual catchments are adopted from the Randolph Glacier Inventory V.6 (RGI)[27] and RGI attributes such as glacier names and IDs are included.

For most timesteps and glaciers several summer acquisitions are available. To identify the most accurate glacier outline, all repeat coverage polygons are stacked individually for each glacier and polygons are selected automatically. Therefore, a set of criteria is evaluated for each glacier with several acquisitions available to derive the most reasonable glacier outline. As spatial parameters, the extent and area of the newly created outline is compared to the respective reference glacier statistics (of RGI) to identify exceptionally large, and thus unlikely, changes in glacier area and extent, e.g., due to partially cloud cover or seasonal snow patches. In addition, the Landsat Quality Assessment Band[37] of the respective original image is used to derive a relative measure of so-called "clear" pixels within the direct vicinity of the glacier outline to further identify cloud or snow biased outlines. For glaciers with several available outlines of equal quality, the outline with the smallest respective glacier area is selected. In cases of incomplete spatial coverage of glacier catchments by all available acquisitions, polygon fragments from different acquisitions are combined and thereafter manually checked. In each case, the outline acquisition date(s) are preserved for each glacier. Outlines with a unique acquisition date are then used to calculate an area-weighted median acquisition date for each subregion, based on the glacier area enclosed by the respective outline, to derive regional area change rates during the observation periods.

The resulting glacier inventories are hereafter named according to their temporal composition. The first inventory (inventory 2000) includes primarily images from 2000 (61.3%) and 1999 (35.4%), the second inventory (inventory 2011) was created

entirely from 2011 images and the third inventory (inventory 2014) mainly from 2014 (41.4%) and 2013 (35.0%) acquisitions.

Finally, each new inventory is visually inspected and remaining misclassified areas, such as patches of seasonal snow or proglacial lakes, are corrected manually based on false color composites, Landsat 7 panchromatic bands and elevation change raster. Debris-covered glacier outlines are also manually corrected according to respective elevation change fields and high-resolution satellite images (Google Earth). For the period 2013–2014, debris-covered glacier tongues are additionally compared with coherence estimates[38] of Sentinel 1 image pairs from 2015 to distinguish debris-covered ice from rocks.

**Elevation change calculation**. We compute digital elevation models (DEM) and elevation change rates from Synthetic Aperture Radar data from the Shuttle Radar Topography Mission (SRTM) of the National Aeronautics and Space Administration (NASA) and of the TerraSAR-X add-on for Digital Elevation Measurement mission (TanDEM-X), operated by the German Aerospace Center (DLR) and Astrium Defense and Space. The SRTM dataset provides a consistent C-band DEM which was acquired during 11 days in February 2000 and covers all landmasses between 60°N and 56°S[39]. We use the void-filled LP DAAC NASA Version 3 product with a ground resolution of 1 arcsec[40]. The TanDEM-X mission provides high-resolution X-Band acquisitions from 2010 onward with several complete coverages of the European Alps between 2011 and 2014. Within this study we produce two DEM mosaics from TanDEM-X CoSSC tiles for the periods 2011–2012 and 2013–2014 which are the only periods with enough acquisitions to cover the entire Alps. Whenever possible, we select TanDEM-X acquisitions from the same season as the SRTM DEM to minimize differences due to either radar signal penetration or snow accumulation at the acquisition time which can bias the elevation change measurement.

Each TanDEM-X elevation model is processed by using differential SAR interferometry, according to the workflow described in previous studies[20–23]. Differential interferograms are calculated using the void-filled SRTM DEM as reference surface. Subsequently, each interferogram is filtered and unwrapped by different algorithms (minimum cost flow & brach cut). The best results are selected manually and converted to elevation values by adding the reference surface heights. Thereafter, the newly created TanDEM-X DEMs are geocoded and coregistered to the reference DEM (SRTM DEM) to further reduce deviations between the DEM datasets. Each TanDEM-X DEM is vertically and horizontally coregistered to the reference DEM surface with an iterative process on stable areas. Those stable areas are selected by removing glacierized areas and slopes larger 15°. In addition, densely vegetated areas are excluded by using Landsat vegetation masks (Normalized Difference Vegetation Index). Subsequently, DEM mosaics are obtained from adjacent TanDEM-X DEMs and the acquisition date of each pixel is preserved alongside the elevation value. Finally, the coregistered TanDEM-X mosaic and the reference DEM (non-void-filled SRTM DEM) are differenced and change rates are computed using the respective TanDEM-X acquisition date and the date of the reference DEM. For the SRTM DEM we use the mean date (16-Feb-2000). Data voids due to gaps in the SRTM or TanDEM-X DEMs are filled by applying an elevation change versus altitude function, based on aggregated elevation change rates within 100 m elevation bins[41]. Contrasting to the hypsometric interpolation applied previously[21], we did not apply the three times the normalized median absolute deviation filter, which can introduce a bias on regional scales, particularly in the accumulation zones, as shown by a recent study[42]. By testing of different filter approaches and manual inspection of the revealed results, a 1–99% quantile filter for each elevation bin was chosen instead to remove outliers in the region-wide hypsometric analysis. In addition, we remove steep slopes (>50°) where accumulation is negligible[43]. As elevation reference, we use the void-filled SRTM DEM for the aggregation of elevation bins and hypsometric interpolation.

**Geodetic mass change**. Area and elevation change measurements are converted to mass budgets following the UNESCO definitions for glacier mass change estimates[44]. We calculate geodetic mass changes for the periods 2000–2012 and 2000–2014, using the earliest glacier inventory ($S_1$, 2000) as baseline and the inventories ($S_2$) of 2011 and 2014, respectively, to determine a temporal mean glacier area ($S_3$) for both observation periods as recommended by the UNESCO. Initially, elevation change rates are integrated for both periods over the respective maximum glacier area ($S_{max}$, spatial union of $S_1$ and $S_2$) and the change volume is calculated by multiplication of the derived elevation change rate and $S_{max}$. In addition, we apply a correction for SAR signal surface penetration ($V_{pen}$, see uncertainty section), which leads to an underestimation of volume change by the relative difference in signal penetration of the X- and C-band SAR. This bias volume due to signal penetration is then added to the measured volume change to derive the full glacier volume change. Subsequently, mass-change rates are estimated by applying a conversion factor assuming a mean density of 850 ± 60 kg m$^{-3}$ based on a study of alpine glaciers[45]. We did not apply a variable density conversion as equilibrium line altitudes during the observation period are only available for a limited number of glaciers from a few regions in the Alps. Accumulation areas in many subregions are also very small and similar to the highest glacierized elevations. Therefore, we decided to use a

constant conversion factor to provide a better comparability to other studies on geodetic mass change.

Finally, we determine specific-mass-change rates by dividing the volume changes by the respective temporal mean glacier area ($S_3$) and multiplying with the density conversion factor.

**Uncertainty assessment of geodetic mass change.** We calculate the geodetic mass change uncertainty according to Eq. (1) with $\Delta M/\Delta t$ being the mass change estimate, $\Delta h/\Delta t$ the average elevation change rate on the whole glacier surface, $S_1$ and $S_2$ the respective glacier areas at beginning and end of the observation period and $p$ the applied volume to mass conversion factor:

$$\delta_{\Delta M/\Delta t} = \sqrt{\left(\frac{\Delta M}{\Delta t}\right)^2 \times \left\{ \left(\frac{\delta_{\Delta h/\Delta t}}{\frac{\Delta h}{\Delta t}}\right)^2 + \left(\frac{\delta_{S_1}}{S_1}\right)^2 + \left(\frac{\delta_{S_2}}{S_2}\right)^2 + \left(\frac{\delta_p}{p}\right)^2 \right\} + \left(\frac{V_{pen}}{\Delta t} \times p\right)^2}$$

(1)

Which considers the following terms:

Error from the DEM differencing (including spatial autocorrelation and hypsometric gapfilling).
Error from glacier areas (independent errors from $S_1$ and $S_2$).
Uncertainty from volume to mass conversion using a fixed density.
Uncertainty from radar signal penetration.

The relative vertical precision of the elevation changes on the glacier surface and the hypsometric gapfilling contribute to the accuracy of the elevation change measurements ($\delta_{\Delta h/\Delta t}$). Elevation changes $\Delta h/\Delta t$ on stable areas, excluding glaciers, water and dense vegetation, are extracted and strongly deviating values are removed with a 2–98% quantile filter. Then, the elevation change values are aggregated in 5° slope bins (Supplementary Fig. 8) and standard deviations ($\sigma_{\Delta h/\Delta t}$) are computed to account for the dependance between surface slope and $\Delta h/\Delta t$ accuracy.

Each slope bin is filtered (2–98% quantile) to remove remaining artifacts. Eventually, the relative vertical precision of $\Delta h/\Delta t$ on the glacier areas is computed by weighting the obtained offsets for each slope bin by the slope distribution on glacier area ($\sigma_{\Delta h/\Delta t \ AW}$) (Supplementary Table 3). To include the uncertainty contribution by spatial autocorrelation we generated semivariograms of 100,000 random $\Delta h/\Delta t$ samples on stable areas and derive a mean lag distance ($d_l$) of ~312 m. We follow a previous approach[46] and estimate the accuracy of the region-wide average elevation changes according to Eq. (2):

$$S_{cor} = d_l^2 \times \pi$$
$$\delta_{\Delta h/\Delta t} = \sqrt{\frac{S_{cor}}{5 \times S_G}} \times \sigma_{\Delta h/\Delta t \ AW} \quad \text{for } S_G > S_{cor}$$
$$\delta_{\Delta h/\Delta t} = \sigma_{\Delta h/\Delta t \ AW} \quad \text{for } S_G < S_{cor}$$

(2)

where $S_{cor}$ is the correlation area and $S_G$ the mean glacier area ($S_3$) multiplied by the empirical weighting factor 5[46].

To account for errors due to misclassified glacier areas, we refer to a detailed comparison of automatically and manually classified outlines of alpine glaciers[47]. The authors found a deviation in area of 3%, corresponding to a perimeter to area ratio of 5.03 km$^{-1}$. To account for different glacier geometries within our regional subdivisions (perimeter to area ratios ranging from 5.49 to 11.62 km$^{-1}$), we apply a scaling factor to better represent the larger area determination error in regions with very small glaciers. In addition, we estimate the accuracy of each glacier inventory individually to include both the uncertainties of the glacier area at the beginning ($\delta S_1$) and at the end of the respective observation period ($\delta S_2$) in our final error budget:

$$\delta S_1 = \frac{r_{P_1/S_1}}{r_{P/S_{Paul \ et al.}}} \times 0.03$$

(3a)

$$\delta S_2 = \frac{r_{P_2/S_2}}{r_{P/S_{Paul \ et al.}}} \times 0.03$$

(3b)

SAR signal penetration of the glacier surface strongly depends on the prevailing surface conditions during the SRTM and TanDEM-X DEM acquisitions. In general, surface penetration of X- and C-band microwave frequencies occurs more frequently during dry and frozen conditions whereas melting glacier ice leads to almost no penetration. When comparing X- and C-band, the signal penetration depth also differs due to different SAR frequencies. On cold dry ice, C-band frequencies can show maximum penetration depths of ~10 m[48,49] while X-band penetration is smaller[50–52]. Due to the unknown surface conditions and varying penetration depths of our DEM acquisitions we cannot quantify a specific penetration value for each DEM. Thus we estimate a bias volume ($V_{pen}$) due to the relative penetration difference of X- to C-band and correct the measured change volume over the periods 2000–2012 and 2000–2014 accordingly. We assume an increasing penetration bias from low to high altitudes. The SRTM DEM and most of the TanDEM-X scenes were acquired during winter months with temperatures well below 0 °C at high altitudes and snow covered frozen glacier surfaces. At low altitudes, in the ablation zones,

penetration depth was probably smaller because the surface ice was closer to the melting point[53]. Therefore, we calculate a linear increase in signal penetration difference from 0 m at minimum glacier elevation to 5 m at maximum glacier elevation for each region as an approximation of the penetration depth difference between C- and X-band[20,21]. We use the 10% percentile of glaciated elevations, according to the glacierized area in 2000, as approximate minimum glacier elevation in each region. Eventually, we integrate $V_{pen}$ in the error budget to account for a potentially under- or overestimated penetration bias in some regions.

**Glacier-specific mass changes and comparison to glaciological measurements.** In addition to the regional measurements, we calculate elevation changes for individual glaciers with glaciological mass balances for the periods 2000–2012 and 2000–2014. We aggregate elevation change rates within elevations bins of 10% of the glacier elevation range or 50 m for glaciers with elevation ranges <500 m or >500 m, respectively[41]. Outliers in each elevation bin are removed by applying a 1–99% quantile filter and data gaps in the hypsometric distribution of elevation changes are filled by a 3rd-order polynomial fit. To account for differences in signal penetration at the glacier surface, we apply on each glacier the regional correction function of the respective subregion. For the specific mass change, we use the temporal mean glacier area and a constant density of 850 kg m$^{-3}$ as for the regional mass changes. To estimate the uncertainty ranges of the individual glacier elevation changes, we use the respective regional elevation change uncertainties.

To compare our geodetic values with glaciological records, we calculate average annual mass changes of 25 glaciers[29] with continuous measurements. For the glaciological averages we use values from the hydrological years 1999/00–2010/11 and 1999/00–2012/13.

**Comparison of present mass-change rates to remaining ice volumes.** To estimate the regional glacier ice volumes in future decades, we use modeled ice thicknesses of all glaciers in the Alps from a recent publication[28]. The glacier-specific thickness raster is aggregated within our regional subdivisions to estimate the present ice volume (2000) of each subregion. Thereafter, the relative regional annual volume change rate of the full observation period 2000–2014 is iteratively subtracted from the respective regional glacier volume (2000) to derive the remaining glacier ice volumes for 2050 and 2100.

## Data availability

Elevation change raster are provided in two versions for both observation periods (2000–2012 & 2000–2014): masked to outlines of the Randolph Glacier Inventory V6.0[27] and to the spatial union area of the respective inventories which were used in this study. In addition, raster masks with the TanDEM-X acquisition dates are included to derive the exact observation period of each cell. Elevation change maps and date masks are provided via the World Data Center PANGAEA (https://www.pangaea.de/) operated by AWI Bremerhaven at https://doi.pangaea.de/10.1594/PANGAEA.914118. Glacier-specific geodetic mass change estimates are available via the World Glacier Monitoring Service (WGMS: https://wgms.ch/) and glacier outlines via the Global Land Ice Measurements from Space (GLIMS: https://www.glims.org/).

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

## Acknowledgements

This study was financially supported by the grant BR2105/14-1 within the DFG Priority Program "Regional Sea Level Change and Society", the Emerging Fields Initiative (EFI) of the Friedrich-Alexander University Erlangen-Nürnberg and by the Bundesministerium für Wirtschaft und Energie (grant no. 50EE1544 GEKKO). We thank the World Glacier Monitoring Service (WGMS) and the Global Land Ice Measurements from Space (GLIMS) communities for free and open access to their datasets and the Copernicus Climate Change Service (C3S) which is implemented by the European Centre for Medium-range Weather Forecasts (ECMWF) on behalf of the European Commission.

## Author contributions

C.S. processed the area and elevation change data, created the graphs and wrote the paper. The analysis code was jointly developed by C.S., P.M., and T.C.S. S.L. contributed outlines of debris-covered glaciers. M.Z. contributed to the comparison of present mass-change rates with remaining ice volumes. M.H.B. initiated and led the study. All authors revised the paper.

## Competing interests

The authors declare no competing interests.
