## [Peer Review File · Nature Communications]

Reviewers' comments:

Reviewer #1 (Remarks to the Author):

Review of ‘Rapid 21st century glacier retreat and downwasting throughout the European Alps’ by Sommer et al.

Under review for *Nature Communications*

In this manuscript, Sommer and colleagues present the results from the first region-wide consistent estimate of elevation and mass change over glaciers in the European Alps for the period 2000-2012/14. They do this by comparing the surface elevation over glaciers from a widely used DEM for 2000 (Shuttle Radar Topographic Mission - SRTM) with surface elevations in 2012/14 derived from the TerraSAR-X-Add-On for Digital Elevation measurements mission (TanDEM-X). For the period 2000-2012/14, they find that surface elevation changes over glaciers typically vary between -0.5 and -0.9 m w.e. a⁻¹, with in some cases mass losses exceeding 1 m w.e. a⁻¹.

This is an interesting and carefully presented study with clear figures. The geodetic mass balances put forward by the authors are unique as they are the first region-wide consistent elevation changes over the European Alps. They will be of large relevance for scientists working on cryospheric topics and in related fields. Geodetic mass balance estimates at the individual glacier level are for instance used to calibrate and evaluate the performance of glacier evolution models. As such, these estimates directly influence future estimates of glacier changes in the European Alps. The work fits in a series of studies on glacier mass changes conducted in the group of M. Braun (at the University of Erlangen), which has so far mostly focused on South America (Malz et al., 2018; Braun et al., 2019; Seehaus et al., 2019). The presented methodology has thus been tested and evaluated in different settings. The applied methodology as such is thus not highly novel, but given the importance of the presented results (as elaborated above), a publication in a leading journal with a broad public, such as *Nature Communications*, seems justified.

I have formulated a list of comments that may seem long at first, but most changes should be relatively easy to implement. I am confident that the authors will be able to tackle the few issues raised here and I hope this will help further improving the clarity and significance of this interesting manuscript.

Major comments

1. **Focus on observed vs. future glacier changes.** When reading the title of the manuscript, I was expecting a manuscript that would focus both on the past and future evolution of glaciers in the European Alps. In reality, >95% of the manuscript deals with observed changes (which is fine!) and the future evolution is addressed very briefly with a very basic approach in which observed changes are projected to the future at a regional scale. I think it is fine to have the future evolution in, and it is interesting to see that the projected changes are close to those from complex approaches (although this may be for the ‘wrong reasons’ and will likely not work in other regions), but it should not feature in the title. The title should refer to the observation period 2000-2014 or something alike (early 21st century,...etc). If the authors want to focus on the future evolution, a far more detailed and relevant analyses would be required, in which for instance results from a glacier evolution model would be calibrated with ‘old’ geodetic mass balance estimates (i.e. prior to this study) and with the ‘new’ geodetic mass balances (from this study) and then compare what the effect is on the modelled future evolution. This is beyond the scope of this study (except if the authors want to undertake this rather major undertaking of course), and should therefore not be part of the title.
2. The **figures**, like most of the text, are very well-presented and easy to follow. It is however a pity that for many figures one also needs to read the caption to understand the meaning of the figure (i.e. the figure cannot be read as a stand-alone). In most figures the reference period (i.e. over which time period are the changes considered) is not given for instance. Would be nice if authors could rework the figures and their labels in order to be interpreted independently. See specific comments below.
3. The data you present in this study will be very interesting for other scientists to use. It is not entirely clear **how you will provide this data?** Ideally this would be at the individual glacier level, where the geodetic mass balance is provided for every glacier you consider and where you directly couple this to Randolph Glacier Inventory (RGI) ID of the glacier. Otherwise this data will require quite a lot of processing before it can be used for most applications (e.g. to calibrate a region-wide model glacier evolution at the individual glacier level to your observed geodetic mass balances). Will you, additionally, also provide the raw data, which could be used to identify the spatial variability in elevation changes within a glacier?

Specific comments

Abstract:

l.14: ‘...even without further greenhouse gas emissions’: this is true and has been shown through model simulations at a global scale and for the European Alps (Marzeion et al., 2018; Zekollari et al., 2020). However, this is not really part of your story (which focuses on observed changes) and does not appear in the main text (also not in the introduction). This could be removed, or otherwise it should also be mentioned in the main text.

l.14-15: ‘Total wastage of glaciers in some regions is expected’: OK, this is also true, but do not really see added value of having this here in abstract. If want to have statement on this, would make sense to relate it to your work through the link between observed changes and modelled future evolution (as the former are typically used for model calibration): see also general comment 1.

l.18: ‘...with a spatial coverage of 86% of all glacierized areas’: if you want to make a stronger case, could also express this in terms of glacier volume covered – which you can directly derive from the consensus ice thickness estimate from Farinotti (2019). Guess you will end up somewhere around 95% of the total volume of glaciers in the European Alps.

l.20: refer to regions with highest loss (Glarus and Lepontine): OK, but many readers will never have heard of these and not known where this is. Maybe give a more general finding (strongest downwasting in low-elevated regions). Think it would also be nice if you could give some results per country (here and/or in the main text): e.g. *the highest observed mass loss occurs in the Austrian Alps (number), while XX limits the glacier loss in the Swiss Alps (number)* – just made this up, may be the other way round of course. Will in general be relevant to give some results per country as in the wide audience you target many scientists will consider the Alps in terms of separate countries for their research.

l.21: mass loss you give here for the period 2000-2014: to how much of the total ice mass does this approx. correspond?

l.21-23: future evolution: can leave this, but as explained before: may just be lucky that this is in line with other estimates (see e.g. Hock et al., 2019) and likely not the case to work for other regions. Think this adds little to your story and that it would be better to focus on the observations rather than on this very simple extrapolation of current trends (which is in general not a good idea to do in geosciences...).

In general: think that you should highlight what is new in your study more in the abstract! i.e. that it is the first region-wide consistent estimate of geodetic mass balance over European Alps. Is your estimate higher-lower than previous estimates, why? Any other study-specific things to note for others (e.g. can this be seen as an important effort to extrapolate to all regions on Earth?). If need more space in your abstract, can reduce the abstract length by removing some redundant statements (see suggestions above).

Introduction:

l.28-29: glacier length changes since LIA: suggest also referring to glacier length compilation effort by Leclercq and colleagues (2014).

l.31: ‘within the early 21st century’: could you be more specific here? With early 21st century, do you refer to period shorter than the one you consider (e.g. 2000-2003) or is this something similar or even longer (e.g. 2000-2019)?

l.33: ‘water towers’: would make sense to refer to new study by Immerzeel et al. (2020) for this.

l.34-35: references 11-14 for hydrology: you mostly refer to studies not in the European Alps and studies that are already quite old. Could refer to more recent work on this (e.g. Hanzer et al., 2018; Brunner et al., 2019).

l.43: ‘predict water availability’: do you refer to future water availability here?

l.43: you state that for predictions on water availability ‘information on glacier changes are essential’. How is this the case? This is because the ‘information on glacier changes’ (the observation) will in many cases be used to calibrate a glacier evolution model, which will in turn affect the modelled/simulated future glacier evolution, which influences the future runoff (e.g. Brunner et al., 2019). Think you need to specify this, as in current description the link between the present-day observation (on glacier change) and future changes in hydrology is otherwise a bit vague.

l.44: ‘most studies in the Alps’: which studies are you referring to here? Glacier change studies I suppose?

l.46-47: ‘...glacier changes over the entire Alps in the last decade is so far missing’: indeed, and this makes your study an important one! Your study does however, unfortunately not really consider ‘the last decade’ (due to TanDEM-X time coverage I guess): so would reformulate this.

l.49-51: got lost in this sentence. Consider splitting up in two sentences - one on DEMs and one on glacier extents - and to reformulate this.

l.53: period 2000-2012 and 2000-2014: almost the same. Why this period? And is it not possible to do this for other (sub)periods (e.g. to obtain trend)? Most likely not and related to TanDEM-X data, but would be good if you could shortly comment on this here in text.

1.52-54: see comment on 1.43: do not think it is entirely clear what link is between observed glacier changes and modelled future runoff changes (link = glacier modelling that needs to be calibrated with present-day observations and which is then linked to hydrological modelling): provide this link and consider whether information is eventually redundant here or in 1.43, as this boils down to the same. Could also formulate it differently here, and say that uncertainties in future projections are large when you cannot evaluate and/or calibrate model with observations on past changes.

Remotely sensed glacier changes:

- 1.57-59: year is mentioned for SRTM. What about TanDEM-X? Only possible for 2012 and 2014 over Alps? Answer may be common knowledge for glacier remote sensing specialist, but not for everyone in the broad audience you are targeting.
- 1.61: you mention the temporal variations in elevation change here, but unfortunately you cannot say a lot about this. May be a too strong claim that you analyse the temporal variation, as 2000-2012 vs. 2000-2014 is almost the same... No possible to consider other time periods? This question relates to question on 1.53 and 1.57-59.
- 1.64: you refer to previous studies for the technique and details can be found in the methods section, which is perfectly fine. Would however be nice if could a very short insight in what the main challenges of used methodology are (e.g. signal penetration) and what implications are: in 1-2 sentences or so, not forcing the reader to look this up in other papers or in methods.
- 1.68: suggest adding a reference to corresponding figure (1b) here.
- 1.73: ‘...3500 m a.s.l. (Graian, Pennine and Bernese Alps).’: when reading the sentence at first, it seems like balanced conditions are found everywhere >3500. But is only the case in these three regions in fact. Would remove the brackets and change to ‘are observed above 3500 m a.s.l. for the Graian, the Pennine and the Bernese Alps’: no ambiguity then.
- 1.74: no region having significant values → maybe stress that this is at all elevations.
- 1.75: ‘...thinning even in the accumulation areas’: well, in many cases the highest glacier points are not even accumulation areas anymore and the entire glacier is now an ablation area. Could reformulate the sentence, or add this information as an extra sentence.
- 1.76-77: surface lowering up to 5 m a⁻¹ for some regions. Ok, but this is because you consider regional values over elevation bands. In some extreme cases, surface lowering can be in the order of -10 m w.e. a⁻¹: frontal areas with nearly ‘dead’ ice, where elevation change corresponds to the local SMB. Could you mention this and give some idea about how extreme the change can be? Point to be considered here:
 - The lowest glaciers are experiencing the strongest losses: between 2000 and 2012/14 these areas will deglacierate, resulting in smaller elevation changes: e.g. if have an ice thickness loss at a rate of 10 m a⁻¹ and this location becomes ice-free in 2007 → will have mean thickness change of 5 m a⁻¹ over period 2000-2014. How do you deal with a deglaciation during the period 2000-2014? Is probably accounted for implicitly at the glacier level by taking mean glacier area over the two periods, but not sure when determining the local ice thickness changes (which can be >5 m a⁻¹).
- 1.78: ‘highest area reductions’: are these the absolute or the relative reductions? If the former, then this is strongly related to the total area at present of course.
- 1.82: ‘both periods’: not clear which periods these are at this point in the manuscript: specify or simply remove.
- 1.82-87: mass losses are given in m w.e. a⁻¹, but no uncertainty is given here (e.g. due to density assumptions, signal penetration,...). Later on this is done (e.g. 1.92-93): why not here?

Comparison with glaciological measurements:

- 1.117: ‘glaciological mass balance is similar or slightly more negative’: compared to what? Compared to the geodetic mass balances you obtain?
- 1.118: ‘the geodetic measurements’: which ones? The ones you perform in your study from my understanding? Would be good to specify if so.
- 1.119-120: substantial difference between geodetic and glaciological mass balance for five glacier specifically. Can you give a hint where this discrepancy arises? Error in glaciological and/or geodetic? Large uncertainties on the values?
- 1.122-123: ‘Except for the...for all regions’: strange sentence here, would suggest removing here. Would also start a new paragraph at ‘For the Pennine,...’ (1.123): because you here switch to a different kind of analyses: from (i) comparing geodetic vs. glaciological for individual glaciers to (ii) comparing how representative values for individual glaciers are for the region in which they are located.
- 1.125: ‘...which are’: could be more specific here to be clear ‘..., where the glaciological measurements are’

Comparison with other geodetic observations:

- In general: the numbers you give are difficult to compare with those from other studies, as other time periods are considered (e.g. l.141-144, l.156-158). You explain this and provide elements that could explain the differences, but this makes this section rather qualitative. Would potentially be an option to reduce the comparisons with other studies in the main text and provide these in part in the supplementary material. Could then refer to this in the main text in a more compact way, and would open up some space for other analyses to be presented in the main text: e.g. a bit more details about the link of observed changes vs. glacier characteristics, providing some information on the country-level (see comment on l.20),...etc.
- l.136: 'From 1985-1999' → 'For the period 1985-1999'
- l.147-148: '...measurement of -0.89 ± 0.01 m w.e. a⁻¹': maybe better to refer to yours as 'estimate', as you do not refer to the others as a measurement either it seems. Furthermore, very surprising to have an error that is that low.. Just by converting the elevation change with a fixed density assumption I would expect you to have a much larger uncertainty...And need to account for signal penetration,...etc also. Is the error at the individual glacier level always that small?
- l.155: 'relatively low DEM coverage (<70%): in your study?'
- l.159-162: your estimate vs. estimate based on 50 Swiss glaciers: not clear what is being compared here? Is the result from the literature extrapolated to the entire Alps (i.e. are you comparing values over the entire Alps) or are you only considering these 50 glaciers when comparing the results?'
- l.172-181: not really part of the 'comparison with other geodetic observations' section. For this part of the text: refer to major comment 1.

Data & methods: clear and well presented, with a lot of details.

- Not sure how you end up with such low numbers when estimating the uncertainty in your geodetic mass balances (see comment on l.147-148).
- l.406: assuming constant density of 850 kg m^{-3} : can this not be improved in a relatively simple way? For instance by assuming a different conversion rate for the ablation area (ice) and the accumulation area (snow, firn)? Maybe not be as easy as it sounds, or does not make sense, but would be good if you could then explain this in 1-2 sentences.
- l.414-419: in general: a very rough approach. Interesting to see that you end up with volume changes close to those from more complex methods relying on glacier evolution models, but is very minor part of your story and therefore better to not have this in your title: see general comment 1 and related comments.
- l.415-416: '...raster are aggregated' → '...raster is aggregated'
- For the present-day ice volume the year 2000 is mentioned, but the glacier outlines are mainly from 2003 in the RGI over European Alps (Paul et al., 2011; RGI Consortium, 2017). Is this a problem / may this cause a discrepancy in your approach?

Data and materials availability: see general comment 3.

Figure 1:

- Would be nice if you could add years over which the changes are considered directly in the figure. Allows reading the figure without having to refer to the caption.
- In panel c you show the glacier area (for which year?) and the glacier mass changes in the same figure. Is a bit counterintuitive. Would it be possible to have the glacier area in the same panel as the area change (i.e. move to panel b). In such a way, the relative loss could be shown directly (using the same scale / bullet size for the area and for the area change).

Figure 2:

- 'Glacier area': based on what?
- 'Measured area': how is this difference from the 'glacier area'? Change in area between both? Probably better if can refer to specific years for both
- In caption, l.548: 'axis' → 'axes'

Figure 3:

- Black circle, triangle and dotted line: indicate that this refers to data from glaciological measurements (in the figure itself)

Table 1: very nice to have this! Could maybe also consider having some information divided per country here?

References

- Braun, M. H., Malz, P., Sommer, C., Farias-Barahona, D., Sauter, T., Casassa, G., et al. (2019). Constraining glacier elevation and mass changes in South America. *Nature Climate Change*, *9*, 130–136. <https://doi.org/10.1038/s41558-018-0375-7>
- Brunner, M. I., Farinotti, D., Zekollari, H., Huss, M., & Zappa, M. (2019). Future shifts in extreme flow regimes in Alpine regions. *Hydrology and Earth System Sciences*, *23*, 4471–4489. <https://doi.org/10.5194/hess-23-4471-2019>
- Farinotti, D., Huss, M., Fürst, J. J., Landmann, J., Machguth, H., Maussion, F., & Pandit, A. (2019). A consensus estimate for the ice thickness distribution of all glaciers on Earth. *Nature Geoscience*. <https://doi.org/10.1038/s41561-019-0300-3>
- Hanzer, F., Förster, K., Nemeč, J., & Strasser, U. (2018). Projected cryospheric and hydrological impacts of 21st century climate change in the Ötztal Alps (Austria) simulated using a physically based approach. *Hydrology and Earth System Sciences*, *22*, 1593–1614. <https://doi.org/10.5194/hess-22-1593-2018>
- Hock, R., Bliss, A., Marzeion, B., Giesen, R. H., Hirabayashi, Y., Huss, M., et al. (2019). GlacierMIP – A model intercomparison of global-scale glacier mass-balance models and projections. *Journal of Glaciology*. <https://doi.org/10.1017/jog.2019.22>
- Immerzeel, W. W., Lutz, A. F., Andrade, M., Bahl, A., Biemans, H., Bolch, T., et al. (2020). Importance and vulnerability of the world's water towers. *Nature*, *577*, 364–369. <https://doi.org/10.1038/s41586-019-1822-y>
- Leclercq, P. W., Oerlemans, J., Basagic, H. J., Bushueva, I., Cook, A. J., & Le Bris, R. (2014). A data set of worldwide glacier length fluctuations. *The Cryosphere*, *8*(2), 659–672. <https://doi.org/10.5194/tc-8-659-2014>
- Malz, P., Meier, W., Casassa, G., Jaña, R., Skvarca, P., & Braun, M. H. (2018). Elevation and mass changes of the southern Patagonia icefield derived from TanDEM-X and SRTM data. *Remote Sensing*, *10*(2), 1–17. <https://doi.org/10.3390/rs10020188>
- Marzeion, B., Kaser, G., Maussion, F., & Champollion, N. (2018). Limited influence of climate change mitigation on short-term glacier mass loss. *Nature Climate Change*, *8*, 305–308. <https://doi.org/10.1038/s41558-018-0093-1>
- Paul, F., Frey, H., & Bris, R. L. E. (2011). A new glacier inventory for the European Alps from Landsat TM scenes of 2003: challenges and results. *Annals of Glaciology*, *52*(59), 144–152. <https://doi.org/10.3189/172756411799096295>
- RGI Consortium, . (2017). *Randolph Glacier Inventory – A Dataset of Global Glacier Outlines: Version 6.0: Technical Report, Global Land Ice Measurements from Space, Colorado, USA. Digital Media.* <https://doi.org/10.7265/N5-RGI-60>
- Seehaus, T., Malz, P., Sommer, C., Lippl, S., Cochachin, A., & Braun, M. (2019). Changes of the tropical glaciers throughout Peru between 2000 and 2016 – mass balance and area fluctuations. *The Cryosphere*, *13*, 2537–2556.
- Zekollari, H., Huss, M., & Farinotti, D. (2020). On the imbalance and response time of glaciers in the European Alps. *Geophysical Research Letters*, *47*. <https://doi.org/10.1029/2019GL085578>

Reviewer #2 (Remarks to the Author):

General comments

In their manuscript Sommer et al. present glacier changes over the entire Alps for the first decade of the 21st century. They provide a comprehensive data set on glacier-area and regional mass changes based on DEM from radar interferometry in combination with optical satellite images. Despite the uncertainty of the used methods for surveying glacier in steep mountain terrain, the results constitute a methodologically consistent cross-border analysis for a 12-year and a 14-year period. The results are presented along findings of regional glacier change studies and in many cases are in consensus to local mass balance data. Thus the analysis on the scale of the Alps not necessarily presents new results on local change rates, but enables the comparison of recent glacier change observations in different regions for a consistent time period for the first time. This builds the basis for setting local mass balance observations into an Alpine-wide context.

The used remotely sensed data are applicable for calculating glacier changes in a regional analysis, but do not entirely cover all processes of glacier wastage in its detail. The calculated area and elevation changes of a large number of small to very small glaciers and low-lying glacier tongues of mid-size to large glaciers are accompanied with high uncertainties caused by debris cover, steep terrain and topographic shading. Thus, the authors publish regional mean values of glacier change rates and present a sound calculation of uncertainties for the specific mass changes. However, the basic uncertainty of the calculated elevation changes appears to be rather low in relation to the methods applied to the complex terrain of the Alps.

Accordingly, observed processes of glacier wastage such as increasing debris cover and glacier disintegration are not considered for future glacier projection. This implements a high uncertainty of the projected glacier coverage either in percentage or time.

In general the manuscript is concisely written and the results are appropriately discussed in the context of previous literature. Considering the supplementary material the presentation of the results in figures and tables is substantial. Just the results section should be realigned for better comprehensibility.

Sommer et al. provide sufficient methodological detail such that the experiments could be reproduced. Nevertheless, I recommend the authors to publish the glacier outlines as an open-access data set along with the manuscript. Since a number of methods were applied to different optical or elevation data in the recent years, the set of outlines will provide an idea of uncertainties and differences of semi-automatic and manual mapping based on different data sources.

Since no fundamental analysis has to be applied for revisions, I recommend to accept after consideration of the revisions suggested in the specific and detailed comments.

Specific comments

Despite the well elaborated presentation of the methods and the results, there are three main concerns to be considered for revisions.

1 Uncertainty of glacier elevation changes

The presented uncertainties of mean elevation change rates (Tab. 1) appear to be rather low. For both periods they are less than 2% or a maximum of 0.21m for the total period. These are theoretical values regarding co-registration performance in stable terrain, but do not include the uncertainty caused by errors of the DEM in steep terrain, along steep slopes at glacier margins or glacier areas shaded by topography, which is important in particular for small to very small glaciers and low-lying glacier tongues. The presented uncertainty of the elevation change rates is distinctly lower than uncertainties presented in the publications referred to in line 290 (#17-19). For an additional uncertainty estimation, the authors should present the two-year elevation change rates between the two TanDEM-X DEM, as e.g. shown in Malz et al. 2018 (#17). Relating to Tab.1, mean elevation change rates of the regions are varying to be either more or less negative for the

different periods. Accumulating the elevation change rates to total elevation changes of the periods and dividing the difference of the total changes by two years results in elevation change rates varying significantly for nearby regions between 2012 and 2014 (e.g. -0.94 ma^{-1} for region 01 and $+0.18 \text{ ma}^{-1}$ for region 02). In particular, for region 08, the two-year elevation change rate would be -0.85 ma^{-1} and, thus, more negative than the 12/14-year period mean. However, 2013 glacier mass balances have been comparatively less negative compared to mean mass balances since 2000 (see WGM mass balance data, e.g. Hintereisferner and Kesselwandferner). If those differences can be related to the median dates of TanDEM-X acquisitions showing differing snow accumulation, this has to be stated as a potential error source increasing uncertainty of elevation change rates nonetheless.

I suggest to revise the uncertainty estimation of the elevation change (rates). If those uncertainties are found to be higher, uncertainties of (specific) mass change rates have to be corrected accordingly.

2 Future Scenarios

In contrast to existing model simulations of future glacier scenarios of the Alps, the analysis presented here does not consider dynamic adjustments, future scenarios of changing glacier mass balance and already observed processes like increasing debris coverage and glacier disintegration for the calculation of future glacier coverage. The presented scenario is also based on ice thickness data showing high relative deviations in ice thickness particularly for Alpine glaciers. This issues incorporates a high uncertainty of the presented results for 2050 and 2100 either in percentage of glacier coverage or time. The calculated future glacier volume also do not reflect the high variability of recent glacier changes and geometries in each of the regions. Model studies of e.g. Zekollari et al. 2019 (#4) present a large range of results with respect to different scenarios. To my opinion the results presented here are neither necessarily needed for the manuscript, which mainly aims in the presentation of a temporally and regionally consistent analysis of recent change rates, neither are they substantial considering all the assumptions and limitations. Thus, I suggest to remove this part of the results.

3 Results sections

In general, the results (L70-105ff) are hard to read in the present form. I would suggest to realign this section by presenting at least one characteristic result for each region in order of the IDs (West to East) first and then to close with the main findings of the comparison (e.g. L72ff, L77ff).

Detailed comments

L14: Replace 'further' by 'increasing'

L15/16: Revise to 'in glacier area, surface elevation and ice mass'

L21ff: Remove sentence in accordance to specific comments

L29: LIA (1850). Here you may add also literature of observations from Austria

Fischer, A., Seiser, B., Stocker Waldhuber, M., Mitterer, C., and Abermann, J., 2015. Tracing glacier changes in Austria from the Little Ice Age to the present using a lidar-based high-resolution glacier inventory in Austria, *The Cryosphere*, 9, 753-766, doi:10.5194/tc-9-753-2015.

<http://www.the-cryosphere.net/9/753/2015/>

L30: reported. Add also reference #29 Abermann et al. (2009)

L50/51:relocate the given numbers (178 DEMs.; 185 scenes...) to Line 60ff

L71: Highly negative 'mean' elevation changes are recorded ' for both larger sub regions,' the Western...

L72ff: This appears to be a finding which also can be highlighted in the abstract.

L74: significantly: do you mean (statistically) significant?

L77: Is this a statement or a result? If latter, this can also be an important point for the abstract

L101: 'large' instead of larger

L104: Similar to what?

L140: add '-' before 0.10

L165: Those are areas of small glaciers and prone to higher uncertainties in analysis of surface elevation change (see specific comments)

L172ff: With respect to the specific comments, I would suggest to remove this section

L231ff: The climate classification is, as far as I can see, not further used in the analysis and may be removed.

L238: Replace 'periods' by 'dates'

L326: applying a 'conversion factor assuming a mean density' of...

L413ff: With respect to the specific comments, I would suggest to remove this section

a) Response to reviewer 1

Review of ‘Rapid 21 st century glacier retreat and downwasting throughout the European Alps’ by Sommer et al.

Under review for Nature Communications

In this manuscript, Sommer and colleagues present the results from the first region-wide consistent estimate of elevation and mass change over glaciers in the European Alps for the period 2000-2012/14. They do this by comparing the surface elevation over glaciers from a widely used DEM for 2000 (Shuttle Radar Topographic Mission - SRTM) with surface elevations in 2012/14 derived from the TerraSAR-X-Add-On for Digital Elevation measurements mission (TanDEM-X). For the period 2000-2012/14, they find that surface elevation changes over glaciers typically vary between -0.5 and -0.9 m w.e. a ⁻¹, with in some cases mass losses exceeding 1 m w.e. a ⁻¹.

This is an interesting and carefully presented study with clear figures. The geodetic mass balances put forward by the authors are unique as they are the first region-wide consistent elevation changes over the European Alps. They will be of large relevance for scientists working on cryospheric topics and in related fields. Geodetic mass balance estimates at the individual glacier level are for instance used to calibrate and evaluate the performance of glacier evolution models. As such, these estimates directly influence future estimates of glacier changes in the European Alps. The work fits in a series of studies on glacier mass changes conducted in the group of M. Braun (at the University of Erlangen), which has so far mostly focused on South America (Malz et al., 2018; Braun et al., 2019; Seehaus et al., 2019). The presented methodology has thus been tested and evaluated in different settings. The applied methodology as such is thus not highly novel, but given the importance of the presented results (as elaborated above), a publication in a leading journal with a broad public, such as Nature Communications, seems justified.

I have formulated a list of comments that may seem long at first, but most changes should be relatively easy to implement. I am confident that the authors will be able to tackle the few issues raised here and I hope this will help further improving the clarity and significance of this interesting manuscript.

Firstly we would like to thank the reviewer for the very constructive and positive comments and suggestions. We have adopted as many comments as possible. A point-by-point reply is attached below.

Major comments

Focus on observed vs. future glacier changes. When reading the title of the manuscript, I was expecting a manuscript that would focus both on the past and future evolution of glaciers in the European Alps. In reality, >95% of the manuscript deals with observed changes (which is fine!) and the future evolution is addressed very briefly with a very basic approach in which observed changes are projected to the future at a regional scale. I think it is fine to have the future evolution in, and it is interesting to see that the projected changes are close to those from complex approaches (although this may be for the ‘wrong reasons’ and will likely not work in other regions), but it should not feature in the title. The title should refer to the observation period 2000-2014 or something alike (early 21st century,...etc). If the authors want to focus on the future evolution, a far more detailed and relevant analyses would be required, in which for instance results from a glacier evolution model would be calibrated with ‘old’ geodetic mass balance estimates (i.e. prior to this study) and with the ‘new’ geodetic mass balances (from this study) and then compare what the effect is on the modelled future evolution. This is beyond the scope of this study (except if the authors want to undertake this rather major undertaking of course), and should therefore not be part of the title.

→ We agree that the title may be misleading and changed it to: “Rapid glacier retreat and downwasting throughout the European Alps in the early 21st century”. We also changed parts of the main text to avoid predictions such as “54% remaining volume by 2050” (which we cannot provide) to a more general description. Combining a glacier evolution model would be in fact beyond the scope of this study. Nevertheless we are of the opinion that this “scenario” adds value to the overall results by visualizing the large magnitude of observed glacier retreat in comparison to the approximate remaining glacier volume.

The **figures**, like most of the text, are very well-presented and easy to follow. It is however a pity that for many figures one also needs to read the caption to understand the meaning of the figure (i.e. the figure cannot be read as a stand-alone). In most figures the reference period (i.e. over which time period are the changes considered) is not given for instance. Would be nice if authors could rework the figures and their labels in order to be interpreted independently. See specific comments below.

→ We added the reference periods and extended the figure legends/captions.

The data you present in this study will be very interesting for other scientists to use. It is not entirely clear **how you will provide this data?** Ideally this would be at the individual glacier level, where the geodetic mass balance is provided for every glacier you consider and where you directly couple this to Randolph Glacier Inventory (RGI) ID of the glacier. Otherwise this data will require quite a lot of processing before it can be used for most applications (e.g. to calibrate a region-wide model glacier evolution at the individual glacier level to your observed geodetic mass balances). Will you, additionally, also provide the raw data, which could be used to identify the spatial variability in elevation changes within a glacier?

→ We will provide all datasets used in the analysis:

a) Elevation change raster will be provided in two versions for both observation periods (2000-2012 & 2000-2014): masked to Randolph Glacier Inventory outlines and masked to the spatial union area of the respective start and end inventory which were used in this study. Additionally, raster with the TanDEM-X acquisition dates will be included to derive the exact observation period of each cell. All raster data was already submitted to the open-

access PANGAEA data server (<https://www.pangaea.de/>). We will add a respective DOI as soon as the submitted dataset is processed.

b) The glacier outlines (polygons) of each glacier will be included within the GLIMS/WGMS database with additional glacier-specific data, such as measured average elevation change rate and specific mass change and respective uncertainties. We are currently preparing those datasets. We have added a small paragraph in the manuscript to provide this information also to the reader and potential data user.

Specific comments

Abstract:

- 1.14: ‘...even without further greenhouse gas emissions’: this is true and has been shown through model simulations at a global scale and for the European Alps (Marzeion et al., 2018; Zekollari et al., 2020). However, this is not really part of your story (which focuses on observed changes) and does not appear in the main text (also not in the introduction). This could be removed, or otherwise it should also be mentioned in the main text.
→ removed sentence to save space
- 1.14-15: ‘Total wastage of glaciers in some regions is expected’: OK, this is also true, but do not really see added value of having this here in abstract. If want to have statement on this, would make sense to relate it to your work through the link between observed changes and modelled future evolution (as the former are typically used for model calibration): see also general comment 1
→ removed sentence to save space
- 1.18: ‘...with a spatial coverage of 86% of all glacierized areas’: if you want to make a stronger case, could also express this in terms of glacier volume covered – which you can directly derive from the consensus ice thickness estimate from Farinotti (2019). Guess you will end up somewhere around 95% of the total volume of glaciers in the European Alps.
→ We changed the value from area covered to volume covered (~93%).
- 1.20: refer to regions with highest loss (Glarus and Lepontine): OK, but many readers will never have heard of these and not known where this is. Maybe give a more general finding (strongest downwasting in low-elevated regions). Think it would also be nice if you could give some results per country (here and/or in the main text): e.g. *the highest observed mass loss occurs in the Austrian Alps (number), while XX limits the glacier loss in the Swiss Alps (number)* – just made this up, may be the other way round of course. Will in general be relevant to give some results per country as in the wide audience you target many scientists will consider the Alps in terms of separate countries for their research.
→ We decided to present our results primarily within orographic divisions of mountain ranges as many glacier areas in the Western and Eastern Alps are intersected by country borders. A division by countries can be therefore somehow subjective in those regions. Nevertheless we calculated and added country-level results in the main text and Tab.1 for better comparison. We also added “...in the Swiss Glarus and Lepontine Alps...” as most glaciers in those subregions belong to Switzerland. Fig. 1 illustrates clearly where those regions are located.
- 1.21: mass loss you give here for the period 2000-2014: to how much of the total ice mass does this approx. correspond?
→ Approx. -1.2 %/a, we included this information in the new concluding paragraph
- 1.21-23: future evolution: can leave this, but as explained before: may just be lucky that this is in line with other estimates (see e.g. Hock et al., 2019) and likely not the case to work for other regions. Think this adds little to your story and that it would be better to focus on the observations rather than on this very simple extrapolation of current trends (which is in general not a good idea to do in geosciences...)
→ Agree, removed this information from the abstract and focused on observed changes
- In general: think that you should highlight what is new in your study more in the abstract! i.e. that it is the first region-wide consistent estimate of geodetic mass balance over

European Alps. Is your estimate higher-lower than previous estimates, why? Any other study-specific things to note for others (e.g. can this be seen as an important effort to extrapolate to all regions on Earth?). If need more space in your abstract, can reduce the abstract length by removing some redundant statements (see suggestions above).

→ We removed parts of the introductory sentences and added “Compared to previous studies, our results are similar for the central high-altitude Alps but less negative for the lower mountain ranges.”

Introduction

- 1.28-29: glacier length changes since LIA: suggest also referring to glacier length compilation effort by Leclercq and colleagues (2014).
→ added Reference
- 1.31: ‘within the early 21 st century’: could you be more specific here? With early 21 st century, do you refer to period shorter than the one you consider (e.g. 2000-2003) or is this something similar or even longer (e.g. 2000-2019)?
→ Changed to “during the first 5 years of the 21st century” (Haeberli et al. 2007 refer to 2000-2005)
- 1.33: ‘water towers’: would make sense to refer to new study by Immerzeel et al. (2020) for this.
→ changed reference and added Immerzeel et al. (2020)
- 1.34-35: references 11-14 for hydrology: you mostly refer to studies not in the European Alps and studies that are already quite old. Could refer to more recent work on this (e.g. Hanzer et al., 2018; Brunner et al., 2019).
→ Thank you very much for the more recent references. Added references and removed some of the older studies.
- 1.43: ‘predict water availability’: do you refer to future water availability here?
→ Yes, added “future” to sentence
- 1.43: you state that for predictions on water availability ‘information on glacier changes are essential’. How is this the case? This is because the ‘information on glacier changes’ (the observation) will in many cases be used to calibrate a glacier evolution model, which will in turn affect the modelled/simulated future glacier evolution, which influences the future runoff (e.g. Brunner et al., 2019). Think you need to specify this, as in current description the link between the present-day observation (on glacier change) and future changes in hydrology is otherwise a bit vague.
→ Changed sentence to specify link to runoff projections and added reference
- 1.44: ‘most studies in the Alps’: which studies are you referring to here? Glacier change studies I suppose?
→ Yes, added “glacier change studies”
- 1.46-47: ‘...glacier changes over the entire Alps in the last decade is so far missing’: indeed, and this makes your study an important one! Your study does however, unfortunately not really consider ‘the last decade’ (due to TanDEM-X time coverage I guess): so would reformulate this.
→ Yes, the observation period is constrained by the availability of TanDEM-X acquisitions covering the entire Alps. Replaced “last decade” with “early 21st century”.
- 1.49-51: got lost in this sentence. Consider splitting up in two sentences - one on DEMs and one on glacier extents - and to reformulate this.
→ We changed and simplified this sentence by removing the numbers and dates of optical and radar images (can be found now in short method description 1.59ff).

- 1.53: period 2000-2012 and 2000-2014: almost the same. Why this period? And is it not possible to do this for other (sub)periods (e.g. to obtain trend)? Most likely not and related to TandDEM-X data, but would be good if you could shortly comment on this here in text.
 - Yes, calculating the elevation change of different periods is unfortunately not possible due to the availability of TanDEM-X acquisitions. We added a data availability statement within the methods section (L.330).

- 1.52-54: see comment on 1.43: do not think it is entirely clear what link is between observed glacier changes and modelled future runoff changes (link = glacier modelling that needs to be calibrated with present-day observations and which is then linked to hydrological modelling): provide this link and consider whether information is eventually redundant here or in 1.43, as this boils down to the same. Could also formulate it differently here, and say that uncertainties in future projections are large when you cannot evaluate and/or calibrate model with observations on past changes.
 - We removed this redundant part of the sentence to save some space.

Remotely sensed glacier changes:

- 1.57-59: year is mentioned for SRTM. What about TanDEM-X? Only possible for 2012 and 2014 over Alps? Answer may be common knowledge for glacier remote sensing specialist, but not for everyone in the broad audience you are targeting.
 - We moved all the acquisition dates from the previous chapter to this part to have all information on used datasets in one place. The time periods had to be selected due to the availability of DEMs covering the entire Alps and acquisitions from the winter months to match the date of SRTM. We also added that winter 2011/12 and 2013/14 are the only periods with TanDEM-X coverage of all glaciers in the Alps (see comments above).
- 1.61: you mention the temporal variations in elevation change here, but unfortunately you cannot say a lot about this. May be a too strong claim that you analyse the temporal variation, as 2000-2012 vs. 2000-2014 is almost the same... No possible to consider other time periods? This question relates to question on 1.53 and 1.57-59.
 - Currently those are the only time periods with enough acquisitions from the same season to cover the entire Alps. Removed “temporal” and added explanation in methods section (L.330ff).
- 1.64: you refer to previous studies for the technique and details can be found in the methods section, which is perfectly fine. Would however be nice if could a very short insight in what the main challenges of used methodology are (e.g. signal penetration) and what implications are: in 1-2 sentences or so, not forcing the reader to look this up in other papers or in methods.
 - We added 2 sentences on general advantages (independent from clouds, no sensor oversaturation) and disadvantages (signal penetration) of elevation change measurements by radar sensors.
- 1.68: suggest adding a reference to corresponding figure (1b) here.
 - added reference to figure
- 1.73: ‘...3500 m a.s.l. (Graian, Pennine and Bernese Alps).’: when reading the sentence at first, it seems like balanced conditions are found everywhere >3500. But is only the case in these three regions in fact. Would remove the brackets and change to ‘are observed above 3500 m a.s.l. for the Graian, the Pennine and the Bernese Alps’: no ambiguity then.
 - removed brackets
- 1.74: no region having significant values → maybe stress that this is at all elevations.
 - Added “no region having significantly positive values even at highest glacier elevations”.
- 1.75: ‘...thinning even in the accumulation areas’: well, in many cases the highest glacier points are not even accumulation areas anymore and the entire glacier is now an ablation area. Could reformulate the sentence, or add this information as an extra sentence.
 - Changed sentence to “indicating the loss of former accumulation areas and thinning over the entire glacier”.
- 1.76-77: surface lowering up to 5 m a -1 for some regions. Ok, but this is because you consider regional values over elevation bands. In some extreme cases, surface lowering can be in the order of -10 m w.e.a -1 : frontal areas with nearly ‘dead’ ice, where elevation change corresponds to the local SMB. Could you mention this and give some idea about how extreme the change can be? Point to be considered here:
 - o The lowest glaciers are experiencing the strongest losses: between 2000 and 2012/14 these areas will deglacierate, resulting in smaller elevation changes: e.g. if have an ice thickness loss at a rate of 10 m a -1

and this location becomes ice-free in 2007 à will have mean thickness change of 5 m a⁻¹ over period 2000-2014. How do you deal with a deglaciation during the period 2000-2014? Is probably accounted for implicitly at the glacier level by taking mean glacier area over the two periods, but not sure when determining the local ice thickness changes (which can be >5 m a⁻¹).

→ Yes, this is true. We added “average surface lowering up to...”. In general the change rates of the lowest elevation bands might be underestimated due to the complete removal of glacier ice at this elevations. Fig.2 also indicates this by showing that in some regions the lowest elevation bands are not as negative as the next higher elevations. (Another reason for this observation could be also an insulating debris-cover of the lower glacier tongues.) We cannot really compensate for this effect on the average elevation change rates as we would need precise knowledge of the temporal retreat stages of each glacier in the Alps. However, for the specific mass change the area retreat is included by using the mean glacier area which is more accurate compared to studies using only the area at beginning of the observation period.

→ We also added an example of glacier-specific maximum elevation change rates.

- 1.78: ‘highest area reductions’: are these the absolute or the relative reductions? If the former, then this is strongly related to the total area at present of course.
→ Yes, these are absolute reductions. We made some small changes to clarify this.
- 1.82: ‘both periods’: not clear which periods these are at this point in the manuscript: specify or simply remove.
→ “both periods” referred to 2000-2012 and 2000-2014. We changed the sentence to “The highest mass losses are measured in the Glarus, Lepontine and Rhaetian Alps during 2000-2012 and 2000-2014.” to clarify this.
- 1.82-87: mass losses are given in m w.e. a⁻¹ , but no uncertainty is given here (e.g. due to density assumptions, signal penetration,...). Later on this is done (e.g. 1.92-93): why not here?
→ The values refer to the results of these regions of both periods (2000-2012 & 2000-2014). In order to simplify reading we did not state the exact results of each region and each period in the text but rather specified ranges of specific mass change in those regions. Regional values and uncertainties can be also found in Tab.1. We tried to clarify this by changing the introductory sentence (see comment above).

Comparison with glaciological measurements:

- 1.117: 'glaciological mass balance is similar or slightly more negative': compared to what? Compared to the geodetic mass balances you obtain?
→ Yes, compared to the geodetic mass balance. Added: "than the geodetic mass change"
- 1.118: 'the geodetic measurements': which ones? The ones you perform in your study from my understanding? Would be good to specify if so.
→ Changed sentence to "Overall, the geodetic measurements derived from SRTM and TanDEM-X are less negative than ..."
- 1.119-120: substantial difference between geodetic and glaciological mass balance for five glacier specifically. Can you give a hint where this discrepancy arises? Error in glaciological and/or geodetic? Large uncertainties on the values?
→ Identifying reasons for those discrepancies is difficult as the causes are probably glacier-specific and cannot be generalized. We assume that in some cases the radar measurements might be affected by shadow effects which bias the measurement as three of those glaciers are very small (Ciardoney, Wurtenkees, Fontana Bianca ~ 0.5km²). There might have been also snowfall during the DEM acquisitions at some glaciers. This could be the case for Saint Sorlin which is much more positive than the glaciological average during 2000-2012 but similar during 2000-2014. Eventually, the glaciological measurement might be also biased as indicated in the beginning of the chapter.
We added a sentence referring to potential uncertainties in the geodetic estimate to describe the error sources.
- 1.122-123: 'Except for the...for all regions': strange sentence here, would suggest removing here. Would also start a new paragraph at 'For the Pennine,...' (1.123): because you here switch to a different kind of analyses: from (i) comparing geodetic vs. glaciological for individual glaciers to (ii) comparing how representative values for individual glaciers are for the region in which they are located.
→ Removed sentence and started new paragraph
- 1.125: '...which are': could be more specific here to be clear '..., where the glaciological measurements are'
→ Changed sentence accordingly

Comparison with other geodetic observations:

- In general: the numbers you give are difficult to compare with those from other studies, as other time periods are considered (e.g. l.141-144, l.156-158). You explain this and provide elements that could explain the differences, but this makes this section rather qualitative. Would potentially be an option to reduce the comparisons with other studies in the main text and provide these in part in the supplementary material. Could then refer to this in the main text in a more compact way, and would open up some space for other analyses to be presented in the main text: e.g. a bit more details about the link of observed changes vs. glacier characteristics, providing some information on the country-level (see comment on l.20),...etc.
 - We included country-level estimates to ease the comparison to other studies who provided mass change of Alpine countries.
- l.136: ‘From 1985-1999’ à ‘For the period 1985-1999’
 - Done
- l.147-148: ‘...measurement of -0.89 ± 0.01 m w.e. a -1 ’: maybe better to refer to yours as ‘estimate’, as you do not refer to the others as a measurement either it seems. Furthermore, very surprising to have an error that is that low.. Just by converting the elevation change with a fixed density assumption I would expect you to have a much larger uncertainty...And need to account for signal penetration,...etc also. Is the error at the individual glacier level always that small?
 - We checked the results for the Aletsch Glacier and found an error within the mass change calculation. The actual specific mass change rate for 2000-2012 is -1.04 ± 0.16 m w.e.a $^{-1}$. We corrected the value in the text and replaced “measurement” by “estimate”.
- l.155: ‘relatively low DEM coverage (<70%)’: in your study?
 - Yes, changed sentence to “...relatively low DEM coverage (<70%) by SRTM and TanDEM-X...”
- l.159-162: your estimate vs. estimate based on 50 Swiss glaciers: not clear what is being compared here? Is the result from the literature extrapolated to the entire Alps (i.e. are you comparing values over the entire Alps) or are you only considering these 50 glaciers when comparing the results?
 - M. Huss (2012) used the mass balance records of 50 Swiss glaciers and extrapolated to the entire Alps. The compared mass change values also refer to the entire Alps. We changed the sentence to: “Average specific mass changes of ... were estimated, based on records of 50 Swiss glaciers and extrapolation.”
- l.172-181: not really part of the ‘comparison with other geodetic observations’ section. For this part of the text: refer to major comment 1
 - Moved to results chapter and added new concluding paragraph

Data & methods: clear and well presented, with a lot of details.

- Not sure how you end up with such low numbers when estimating the uncertainty in your geodetic mass balances (see comment on l.147-148).
 - Corrected the values for Aletsch Glacier.
 - In general, the offsets after co-registration are low due to the long observation periods (> 10 years) and topography of the Alps. Compared to other mountain ranges, altitudes in most Alpine regions are rather low with few very steep slopes. Also, the large lateral valleys and surrounding lowlands provide sufficient “stable” ground for the co-registration. The abundance of very dense vegetation (e.g. rainforest) or other “unstable” terrain (e.g. flooding areas, large icefields) further improves the accuracy of the DEM adjustment.

- l.406: assuming constant density of 850 kg m⁻³ : can this not be improved in a relatively simple way? For instance by assuming a different conversion rate for the ablation area (ice) and the accumulation area (snow, firn)? Maybe not be as easy as it sounds, or does not make sense, but would be good if you could then explain this in 1-2 sentences.
 - Initially we planned to apply different conversion factors for the ablation and accumulation area. Unfortunately, records of equilibrium line altitudes are only available for a very small number of glaciers in the central, high-altitude Alps (< 30 glaciers). The average of all available equilibrium line altitudes (between 2000 and 2014) would be ~3.100 m a.s.l., which is equal to the highest glacier elevations in many of the smaller subregions. On the other side, estimating firn lines as indicators of equilibrium line altitudes from (optical) remote sensing data is a very complex and time-consuming approach which was beyond the scope of this study and might be also not possible for some regions of the Alps due to the availability of cloud-free late summer images. Additionally, applying density values (like 0.6 g/cm³) in the accumulation area might be quite arbitrary since they are not supported by any measurements. So we do not really see an improvement also considering that the accumulation areas are meanwhile quite small. Therefore, we decided to apply a constant density conversion factor (which was found for glaciers in the European Alps*) to provide a better comparability to other studies in the European Alps (and elsewhere) as there would be only small changes of the mass change results when using a variable density conversion.
 - We added a respective description to the methods section.
 - *Huss, M. Density assumptions for converting geodetic glacier volume change to mass change. *The Cryosphere* 7, 877–887 (2013).

- l.414-419: in general: a very rough approach. Interesting to see that you end up with volume changes close to those from more complex methods relying on glacier evolution models, but is very minor part of your story and therefore better to not have this in your title: see general comment 1 and related comments.
 - Changed title and text accordingly to focus more on past observations

- l.415-416: ‘...raster are aggregated’ à ‘...raster is aggregated’
 - Done

- For the present-day ice volume the year 2000 is mentioned, but the glacier outlines are mainly from 2003 in the RGI over European Alps (Paul et al., 2011; RGI Consortium, 2017). Is this a problem / may this cause a discrepancy in your approach?
 - Compared to the RGI outlines of 2003 the glacier areas of our 2000 inventory are ~100km² larger (~5%). Based on this larger area the “start” ice volume would be slightly underestimated. However, we do not assume that this difference causes a major discrepancy as the approach itself is very simplistic and rather aims at showing the immediate

vulnerability of smaller glacier regions than providing an exact temporal estimate of regional deglaciation, which is not possible with this approach in any case.

Data and materials availability: see general comment 3.

→ Added statement to methods section

Figure 1:

- Would be nice if you could add years over which the changes are considered directly in the figure. Allows reading the figure without having to refer to the caption.
→ Added years in all figure legends
- In panel c you show the glacier area (for which year?) and the glacier mass changes in the same figure. Is a bit counterintuitive. Would it be possible to have the glacier area in the same panel as the area change (i.e. move to panel b). In such a way, the relative loss could be shown directly (using the same scale /bullet size for the area and for the area change).
→ Unfortunately the total area (2000) pie charts in panel c also show the coverage of SRTM/TanDEM-X (measured/interpolated). By moving the pie charts to panel b it would be probably confusing to have area changes and spatial coverage of elevation change measurements in one figure. However, we replaced the absolute area change (km²) in panel b with area change rates (km²/a) to be more consistent with panel a and c which also show annual change rates.

Figure 2:

- ‘Glacier area’: based on what?
→ This refers to the glacier areas at beginning of the observation period (=2000). Changed figure legend to “Glacier area (2000)”
- ‘Measured area’: how is this difference from the ‘glacier area’? Change in area between both? Probably better if can refer to specific years for both
→ “Measured area” refers to the fraction of the total glacier area (2000) measured by SRTM/TanDEM-X. We changed the figure legend to “Glacier area measured by SRTM/TanDEM-X (2000)” and extended the description in the caption.
- In caption, l.548: ‘axis’ à ‘axes’
→ Corrected

Figure 3:

- Black circle, triangle and dotted line: indicate that this refers to data from glaciological measurements (in the figure itself)
→ Added “glac. MB” to figure legend

Table 1: very nice to have this! Could maybe also consider having some information divided per country here?

- We calculated and added country-level values for the Alpine countries with glacier areas larger 10km².

References

- Braun, M. H., Malz, P., Sommer, C., Farias-Barahona, D., Sauter, T., Casassa, G., et al. (2019). Constraining glacier elevation and mass changes in South America. *Nature Climate Change*, 9, 130–136. <https://doi.org/10.1038/s41558-018-0375-7>
- Brunner, M. I., Farinotti, D., Zekollari, H., Huss, M., & Zappa, M. (2019). Future shifts in extreme flow regimes in Alpine regions. *Hydrology and Earth System Sciences*, 23, 4471–4489. <https://doi.org/10.5194/hess-23-4471-2019>
- Farinotti, D., Huss, M., Fürst, J. J., Landmann, J., Machguth, H., Maussion, F., & Pandit, A. (2019). A consensus estimate for the ice thickness distribution of all glaciers on Earth. *Nature Geoscience*. <https://doi.org/10.1038/s41561-019-0300-3>
- Hanzer, F., Förster, K., Nemeč, J., & Strasser, U. (2018). Projected cryospheric and hydrological impacts of 21st century climate change in the Ötztal Alps (Austria) simulated using a physically based approach. *Hydrology and Earth System Sciences*, 22, 1593–1614. <https://doi.org/10.5194/hess-22-1593-2018>
- Hock, R., Bliss, A., Marzeion, B., Giesen, R. H., Hirabayashi, Y., Huss, M., et al. (2019). GlacierMIP – A model intercomparison of global-scale glacier mass-balance models and projections. *Journal of Glaciology*. <https://doi.org/10.1017/jog.2019.22>
- Immerzeel, W. W., Lutz, A. F., Andrade, M., Bahl, A., Biemans, H., Bolch, T., et al. (2020). Importance and vulnerability of the world's water towers. *Nature*, 577, 364–369. <https://doi.org/10.1038/s41586-019-1822-y>
- Leclercq, P. W., Oerlemans, J., Basagic, H. J., Bushueva, I., Cook, A. J., & Le Bris, R. (2014). A data set of worldwide glacier length fluctuations. *The Cryosphere*, 8(2), 659–672. <https://doi.org/10.5194/tc-8-659-2014>
- Malz, P., Meier, W., Casassa, G., Jaña, R., Skvarca, P., & Braun, M. H. (2018). Elevation and mass changes of the southern Patagonia icefield derived from TanDEM-X and SRTM data. *Remote Sensing*, 10(2), 1–17. <https://doi.org/10.3390/rs10020188>
- Marzeion, B., Kaser, G., Maussion, F., & Champollion, N. (2018). Limited influence of climate change mitigation on short-term glacier mass loss. *Nature Climate Change*, 8, 305–308. <https://doi.org/10.1038/s41558-018-0093-1>
- Paul, F., Frey, H., & Bris, R. L. E. (2011). A new glacier inventory for the European Alps from Landsat TM scenes of 2003: challenges and results. *Annals of Glaciology*, 52(59), 144–152. <https://doi.org/10.3189/172756411799096295>
- RGI Consortium, . (2017). Randolph Glacier Inventory – A Dataset of Global Glacier Outlines: Version 6.0: Technical Report, Global Land Ice Measurements from Space, Colorado, USA. Digital Media. <https://doi.org/10.7265/N5-RGI-60>
- Seehaus, T., Malz, P., Sommer, C., Lippl, S., Cochachin, A., & Braun, M. (2019). Changes of the tropical glaciers throughout Peru between 2000 and 2016 – mass balance and area fluctuations. *The Cryosphere*, 13, 2537–2556.
- Zekollari, H., Huss, M., & Farinotti, D. (2020). On the imbalance and response time of glaciers in the European Alps. *Geophysical Research Letters*, 47. <https://doi.org/10.1029/2019GL085578>

b) Response to reviewer 2

Reviewer #2 (Remarks to the Author):

General comments

In their manuscript Sommer et al. present glacier changes over the entire Alps for the first decade of the 21st century. They provide a comprehensive data set on glacier-area and regional mass changes based on DEM from radar interferometry in combination with optical satellite images. Despite the uncertainty of the used methods for surveying glacier in steep mountain terrain, the results constitute a methodologically consistent cross-border analysis for a 12year and a 14year period. The results are presented along findings of regional glacier change studies and in many cases are in consensus to local mass balance data. Thus the analysis on the scale of the Alps not necessarily presents new results on local change rates, but enables the comparison of recent glacier change observations in different regions for a consistent time period for the first time. This builds the basis for setting local mass balance observations into an Alpine-wide context.

The used remotely sensed data are applicable for calculating glacier changes in a regional analysis, but do not entirely cover all processes of glacier wastage in its detail. The calculated area and elevation changes of a large number of small to very small glaciers and low-lying glacier tongues of mid-size to large glaciers are accompanied with high uncertainties caused by debris cover, steep terrain and topographic shading. Thus, the authors publish regional mean values of glacier change rates and present a sound calculation of uncertainties for the specific mass changes. However, the basic uncertainty of the calculated elevation changes appears to be rather low in relation to the methods applied to the complex terrain of the Alps.

Accordingly, observed processes of glacier wastage such as increasing debris cover and glacier disintegration are not considered for future glacier projection. This implements a high uncertainty of the projected glacier coverage either in percentage or time.

In general the manuscript is concisely written and the results are appropriately discussed in the context of previous literature. Considering the supplementary material the presentation of the results in figures and tables is substantial. Just the results section should be realigned for better comprehensibility.

Sommer et al. provide sufficient methodological detail such that the experiments could be reproduced. Nevertheless, I recommend the authors to publish the glacier outlines as an open-access data set along with the manuscript. Since a number of methods were applied to different optical or elevation data in the recent years, the set of outlines will provide an idea of uncertainties and differences of semi-automatic and manual mapping based on different data sources.

Since no fundamental analysis has to be applied for revisions, I recommend to accept after consideration of the revisions suggested in the specific and detailed comments.

Firstly we would like to thank the reviewer for the very constructive and positive comments and suggestions. We have adopted as many comments as possible. A point-by-point reply is attached below.

Specific comments

Despite the well elaborated presentation of the methods and the results, there are three main concerns to be considered for revisions.

1 Uncertainty of glacier elevation changes

The presented uncertainties of mean elevation change rates (Tab. 1) appear to be rather low. For both periods they are less than 2% or a maximum of 0.21m for the total period. These are theoretical values regarding co-registration performance in stable terrain, but do not include the uncertainty caused by errors of the DEM in steep terrain, along steep slopes at glacier margins or glacier areas shaded by topography, which is important in particular for small to very small glaciers and low-lying glacier tongues. The presented uncertainty of the elevation change rates is distinctly lower than uncertainties presented in the publications referred to in line 290 (#17-19)

→ To account for slope related DEM uncertainties we aggregated offsets between SRTM and TanDEM-X on all non-glacierized pixels within 5° slope bins (shown in Fig.S8) and weighted the offsets of those bins accordingly to the total glacier area of each slope bin. However, concerning the regional elevation change results, the uncertainties are rather low because the largest fraction of glacier areas are part of the large valley glaciers with relatively small slopes (< 35°). Nevertheless, we agree that results of small to very small glaciers are more likely to be biased due to effects of the radar viewing geometry and topography. We therefore added a respective statement in the “Comparison to glaciological measurements” chapter (which includes measurements for very small glaciers by SRTM/TanDEM-X) as we can hardly quantify such effects on very small glaciers.

→ Compared to the respective studies from South America we improved the removal of outlier pixels and overall DEM quality. Also, compared to the South American study sites, the topography of the European Alps allows for more accurate measurements of surface elevation changes due to the much smaller areas and lower elevations in most ranges of the Alps. The surrounding lowlands and large lateral valleys of the Alps provide more than sufficient “stable” terrain which enables a very precise co-registration between SRTM and TanDEM-X. Many regions in South America are more challenging in this regard which leads to larger uncertainties (e.g. steep volcanoes and rainforest in the tropics and steep fjords, large water bodies and islands in Patagonia and Tierra del Fuego). Additionally, we found no regions with large “unstable” areas (such as very dense vegetation or flooding areas) which also improved the DEM adjustment.

For an additional uncertainty estimation, the authors should present the two-year elevation change rates between the two TanDEM-X DEM, as e.g. shown in Malz et al. 2018 (#17).

→ We did not compute elevation change rates between winter 2011/2012 and winter 2013/2014 as the results probably would be heavily biased by seasonal accumulation/ablation and varying surface conditions due to the different TanDEM-X acquisition times. In our experience, DEM differences of less than three years do not provide meaningful change rates as the signal is superimposed by climatic conditions, signal penetration and local co-registration offsets. In the case of Malz et al. (2018) the period between the TanDEM-X acquisitions of the Southern Patagonia Icefield was approximately 4 years. Nevertheless, the hypsometric distribution of elevation changes showed patterns which might have been related to some heavy precipitation events in the meantime or varying signal penetration in the accumulation areas. For the Alps, the temporal difference between regional acquisitions is much shorter and even less than 2 years in some regions.

Relating to Tab.1, mean elevation change rates of the regions are varying to be either more or less negative for the different periods. Accumulating the elevation change rates to total elevation changes of the periods and dividing the difference of the total changes by two years results in elevation change rates varying significantly for nearby regions between 2012 and 2014 (e.g. -0.94 ma⁻¹ for region 01 and +0.18 ma⁻¹ for region 02). In particular, for region 08, the two-year

elevation change rate would be -0.85 ma^{-1} and, thus, more negative than the 12/14-year period mean. However, 2013 glacier mass balances have been comparatively less negative compared to mean mass balances since 2000 (see WGM mass balance data, e.g. Hintereisferner and Kesselwandferner). If those differences can be related to the median dates of TanDEM-X acquisitions showing differing snow accumulation, this has to be stated as a potential error source increasing uncertainty of elevation change rates nonetheless.

→ Yes, such differences between adjacent regions are probably (partially) related to TanDEM-X acquisitions of different months with varying surface conditions. We tried to minimize the temporal offsets between SRTM and TanDEM-X but unfortunately it is not possible for all regions to only use acquisitions from the same month (e.g. February) or with the same prevailing surface conditions. For example, thawing or snowfall at the acquisition time of TanDEM-X can influence the measurement by physically changing the surface height or altering the depth of signal penetration. We added a statement to indicate the acquisition time as an error source in the results chapter (L144-147) and in the methods section (L332-335).

I suggest to revise the uncertainty estimation of the elevation change (rates). If those uncertainties are found to be higher, uncertainties of (specific) mass change rates have to be corrected accordingly.

2 Future Scenarios

In contrast to existing model simulations of future glacier scenarios of the Alps, the analysis presented here does not consider dynamic adjustments, future scenarios of changing glacier mass balance and already observed processes like increasing debris coverage and glacier disintegration for the calculation of future glacier coverage. The presented scenario is also based on ice thickness data showing high relative deviations in ice thickness particularly for Alpine glaciers. This issues incorporates a high uncertainty of the presented results for 2050 and 2100 either in percentage of glacier coverage or time. The calculated future glacier volume also do not reflect the high variability of recent glacier changes and geometries in each of the regions. Model studies of e.g. Zekollari et al. 2019 (#4) present a large range of results with respect to different scenarios. To my opinion the results presented here are neither necessarily needed for the manuscript, which mainly aims in the presentation of a temporally and regionally consistent analysis of recent change rates, neither are they substantial considering all the assumptions and limitations. Thus, I suggest to remove this part of the results.

→ We agree that the presented “future scenario” is based on a very much simplified approach and far less accurate than respective modeling studies. Our aim was not to predict the likely evolution of glacier mass, which we obviously cannot do, but rather show the imminent vulnerability against almost complete deglaciation of some lower Alpine mountain ranges. We therefore think that this “scenario” adds value to the overall results by visualizing the very large magnitude of recent glacier retreat in comparison to the approximate remaining ice volume. Nevertheless, we reformulated the respective text passages to focus more on the measured glacier changes and replaced “2050” and “2100” by “end of the century”. We also made small changes of the title to emphasize our measurement period (“early 21st century”).

3 Results sections

In general, the results (L70-105ff) are hard to read in the present form. I would suggest to realign this section by presenting at least one characteristic result for each region in order of the IDs (West to East) first and then to close with the main findings of the comparison (e.g. L72ff, L77ff).

→ We agree with the reviewer that there are different possibilities to present the results and depending on the persons background and interest a different order might support the reading flow. However, we chose our structure because it follows the order of our figures. Following the reviewers suggestion would require the reader to switch between figures which other readers might

not find intuitive. As a consequence we would like to stay with the proposed order for the presentation of results.

Detailed comments

L14: Replace 'further' by 'increasing'

→ We had to remove this sentence to save space

L15/16: Revise to 'in glacier area, surface elevation and ice mass'

→ Ok, done

L21ff: Remove sentence in accordance to specific comments

→ see answer to specific comment 2

L29: LIA (1850). Here you may add also literature of observations from Austria

Fischer, A., Seiser, B., Stocker Waldhuber, M., Mitterer, C., and Abermann, J., 2015. Tracing glacier changes in Austria from the Little Ice Age to the present using a lidar-based high-resolution glacier inventory in Austria, *The Cryosphere*, 9, 753-766, doi:10.5194/tc-9-753-2015. <http://www.the-cryosphere.net/9/753/2015/>

→ Reference added, Fischer et al. 2015

L30: reported. Add also reference #29 Abermann et al. (2009)

→ Reference added

L50/51:relocate the given numbers (17ß DEMs.; 185 scenes...) to Line 60ff

→ Numbers moved to lines 57ff

L71: Highly negative 'mean' elevation changes are recorded ' for both larger sub regions,' the Western...

→ Ok, added

L72ff: This appears to be a finding which also can be highlighted in the abstract.

→ We decided to place this observation in the results chapter as we would need to remove other information from the abstract

L74: significantly: do you mean (statistically) significant?

→ Elevation changes are positive above 3500 m a.s.l. but only slightly. We intended to describe that there is no region with clearly positive elevation changes considering the uncertainty ranges at this elevation bands.

L77: Is this a statement or a result? If latter, this can also be an important point for the abstract

→ This is a result from the comparison of total glacier areas in each subregion and the respective area and mass changes (as shown in Fig. 1). However, we could not include it in the Abstract due to the maximum number of words (150).

L101: 'large' instead of larger

→ Changed to "large"

L104: Similar to what?

→ Changed sentence to "Area changes show a similar pattern as surface elevation changes ..."

L140: add'-' before 0.10

→ Changed to $-0.10 \pm 0.02 \text{ km}^3 \text{ a}^{-1}$

L165: Those are areas of small glaciers and prone to higher uncertainties in analysis of surface elevation change (see specific comments)

→ Areas with small to very small glaciers are more likely to be affected by uncertainties related to glacier outlines, radar geometry, spatial resolution and other effects. For the glacier area assessment we included this by scaling the glacier outline error by the glacier area to perimeter ratio. We also included this potential error source of very small glacier areas in the “Comparison with glaciological measurements” chapter (L.144-147) to indicate reasons for the discrepancies between glaciological and geodetic measurements of some glaciers.

L172ff: With respect to the specific comments, I would suggest to remove this section

→ see answer to specific comment 2

L231ff: The climate classification is, as far as I can see, not further used in the analysis and may be removed.

→ Removed climate classification

L238: Replace ‘periods’ by ‘dates’

→ done

L326: applying a ‘conversion factor assuming a mean density’ of...

→ done

L413ff: With respect to the specific comments, I would suggest to remove this section

→ see answer to specific comment 2

Reviewers' comments:

Reviewer #1 (Remarks to the Author):

Review revised manuscript ‘Rapid glacier retreat and downwasting throughout the European Alps in the early 21st century’ by Sommer et al.

Under review for *Nature Communications*

It is with great interest that I read the revised manuscript submitted by Sommer and colleagues. The initial work was of high scientific quality and relevance, and for this revised version of the manuscript the authors have addressed the issues that I had raised and incorporated most of the comments. The authors decided to keep the future projections in their storyline, which is fine, but they have removed them from the title, which I think is justified. The availability of the data that the authors describe is elaborate and entirely fits in the ‘open’ philosophy that ‘Nature Communications’ embodies. I am very happy to recommend this work for publication, and have formulated a few very minor comments/suggestions that the authors could still consider including:

- l.20: ‘...our results are similar...’: is a bit vague. Could you be more specific which results? e.g. ‘our estimated mass losses are...’ or something alike.
- l.157: ‘...provide a data basis for...’: a ‘database’?
- l.164: ‘...were found..’: was not clear to me at first if this was referring to your own work or the literature. Seems to be the former. Maybe reformulate to ‘We propose two approaches...’. Moreover, could you be a bit more specific what these two approaches refer to?
- l.168-169: ‘...probably linked to warmer summer temperatures in the 21st century’: ok, could partly explain this, but is maybe a bit too simplistic? Another factor that also plays a role is that glaciers retreat and thus also lose area over this time period. Due to this retreat, they lose their lowest parts, where the mass balance is strongly negative, which increases their mass balance. Do I understand correctly that this will also influence the numbers you put forward? In the end, I think this could be seen as a kind of ‘race’ between the warming on the one hand (further decreasing the mass balance) and the retreat (increasing the mass balance). What you suggest is that - over the time period you consider - the warming is ‘outpacing’ the retreat (see also doi.org/10.1029/2019GL085578, which supports this finding).
- l.188-189: comparing your results to an estimate from the literature. How sensitive is this comparison to the value considered for the snow-ice density? Did you use the same conversion as the one used in the study you are comparing to?
- l.193: ‘...is smaller’ ‘are smaller’?
- l.198: explanation related to temperature change. See comment on l.168-169: glaciers also retreated during time period, which will also influence the mass balance, right?
- l.202-204: ‘strongest contributors are the ablation zones of large valley glaciers...’: indeed. May be worth mentioning that this is related to the fact that these are the glaciers that are most out of balance compared to the present-day climate due to their long response times: i.e. these lower parts of these large valley glaciers are relicts from older, colder time periods.

Reviewer #2 (Remarks to the Author):

In the revised version of their manuscript, Sommer and colleagues have addressed all specific and detailed comments accordingly. They have changed the title to more focus on the observed changes. The authors have retained the future scenario analysis, although the importance of this estimation was reduced. With respect to processes like dynamic adjustment and climate change impacts on glacier mass balance, which are not considered in the approach, the results appear to be less robust compared to more detailed model studies. Conversely, the observed volume changes include processes of glacier vanishing such as basal melt and high mass losses due to the present state of low glacier dynamics in particular for many of the small to medium-size glaciers. Thus, the presentation of the results in Line 118ff, also stating the limitations, appears to be appropriate.

Still, I'm concerned on the presented uncertainty of the surface elevation changes. Coregistration will eliminate a number of systematic errors of surface elevation changes (mean/median deviation, shifts). Nevertheless, the standard deviation of deviations between TandemX DEMs and plain surfaces was found to be in the order of several decimetres with RMSE in the order of meters (e.g. Becek et al. 2016, Wessel et al. 2018), and can be expected to be higher for SRTM DEMs (e.g. Kropáček et al. 2014). For their uncertainty estimation, the authors refer to the study of Rolstad et al. (2009). A fundamental parameter of this geostatistical analysis is the standard deviation of Δz ($\sigma\Delta z$). However, applying EQ. 2 using the presented lag distance of 312 m (L 407) and assuming a $\sigma\Delta z$ of 0.5 m for all glaciers of RGI6.0 Central Europe results in an average uncertainty of elevation changes of 0.4 m (i.e. 0.036 ma⁻¹), and 0.15 m (0.012 ma⁻¹) if weighted for glacier area. This is about 4 times to an order of magnitude higher compared to the uncertainties of surface elevation changes presented in Table 1. I agree that this simple calculation cannot match the results of the detailed analysis considering surface slopes. However, I recommend to present $\sigma\Delta z$ for a better reproducibility of uncertainty estimations in one or in a combination of the following ways:

- (i) In Figure S8, the authors present the normalized median absolute deviation (NMAD), which is an error measure in particular for non-normal error distributions (Höhle and Höhle, 2009). If the distribution of the elevation differences between coregistered SRTM and TandemX DEM in stable areas is non-normal, this may be discussed. If the distribution is normal, $\sigma\Delta z$ can be presented instead of NMAD.
- (ii) An additional Table (in the supplement) presenting the total area, total glacier area, glacier area <35° slope, stable non-glacier areas <35° slope and $\sigma\Delta z$ of these areas for each of the 10 regions
- (iii) An additional gridded data set with Δz of all stable non-glacier/coregistration areas considered for uncertainty estimation not masked to outlines of the RGI6.0 (compare Line 475)

If the uncertainties of surface elevation changes are found to be higher, uncertainties of (specific) mass change rates have to be adjusted accordingly.

Apart from this point, Sommer and colleagues present a well elaborated study of a first region-wide consistent analysis of glacier surface elevation changes over the European Alps worth to be published in Nature Communications. Figures and tables of the supplementary material complete the clear presentation of the results in the main text.

References

Becek, K., Koppe, W., Kutoğlu, Ş.H., 2016. Evaluation of Vertical Accuracy of the WorldDEM™ Using the Runway Method. *Remote Sens.*, 8, 934. <https://doi.org/10.3390/rs8110934>

Höhle, J., Höhle, M., 2009. Accuracy assessment of digital elevation models by means of robust statistical methods. *ISPRS J. Photogramm. Remote Sens.* 64 (4),398–406, <https://doi.org/10.1016/j.isprsjprs.2009.02.003>

Kropáček, J., Neckel, N., Bauder, A., 2014: Estimation of Mass Balance of the Grosser Aletschgletscher, Swiss Alps, from ICESat Laser Altimetry Data and Digital Elevation Models. *Remote Sens.*, 6, 5614-5632. <https://doi.org/10.3390/rs6065614>

Rolstad, C., Haug, T., Denby, B., 2009: Spatially integrated geodetic glacier mass balance and its uncertainty based on geostatistical analysis: application to the western Svartisen ice cap, Norway. *J. Glaciol.* 55, 666–680. <https://doi.org/10.3189/002214309789470950>

Wessel, B., Huber, M., Wohlfart, C., Marschalk, U., Kosmann, D., Roth, A., 2018: Accuracy assessment of the global TanDEM-X Digital Elevation Model with GPS data, *ISPRS Journal of Photogrammetry and Remote Sensing*, 139, 171-182. <https://doi.org/10.1016/j.isprsjprs.2018.02.017>

#Reviewer 1

Review revised manuscript ‘Rapid glacier retreat and downwasting throughout the European Alps in the early 21st century’ by Sommer et al.

Under review for Nature Communications

It is with great interest that I read the revised manuscript submitted by Sommer and colleagues. The initial work was of high scientific quality and relevance, and for this revised version of the manuscript the authors have addressed the issues that I had raised and incorporated most of the comments. The authors decided to keep the future projections in their storyline, which is fine, but they have removed them from the title, which I think is justified. The availability of the data that the authors describe is elaborate and entirely fits in the ‘open’ philosophy that ‘Nature Communications’ embodies. I am very happy to recommend this work for publication, and have formulated a few very minor comments/suggestions that the authors could still consider including:

Thank you very much for the suggestions. We made some small changes to the main text accordingly.

l.20: ‘...,our results are similar...’: is a bit vague. Could you be more specific which results? e.g. ‘our estimated mass losses are...’ or something alike.
→ Changed to “...our estimated mass changes...”

l.157: ‘...provide a data basis for...’: a ‘database’?
→ Ok, changed

l.164: ‘...were found..’: was not clear to me at first if this was referring to your own work or the literature. Seems to be the former. Maybe reformulate to ‘We propose two approaches...’. Moreover, could you be a bit more specific what these two approaches refer to?
→ This actually refers to the cited literature and not our study. Paul & Haeberli (2008) presented mass change results by calculating the arithmetic mean of those individual 786 glaciers with elevation change measurements (method A: -7.0 m w.e) and the mean value of the entire glacier area (method B: -10.95 m w.e.). Method B is more similar to our regional mass change estimates (except the use of a different density conversion). We made small changes in the text to clarify this (changed “were found” to “were reported” and added “of this study” in L.166).

l.168-169: ‘...probably linked to warmer summer temperatures in the 21 st century’: ok, could partly explain this, but is maybe a bit too simplistic? Another factor that also plays a role is that glaciers retreat and thus also lose area over this time period. Due to this retreat, they lose their lowest parts, where the mass balance is strongly negative, which increases their mass balance. Do I understand correctly that this will also influence the numbers you put forward? In the end, I think this could be seen as a kind of ‘race’ between the warming on the one hand (further decreasing the mass balance) and the retreat (increasing the mass balance). What you suggest is that - over the time period you consider – the warming is ‘outpacing’ the retreat (see also doi.org/10.1029/2019GL085578, which supports this finding).

→ Yes, large-scale glacier retreat during the study period can influence geodetic mass change results by underestimating the actual elevation change at the glacier front. This is particularly true for studies which rely only on the approximate glacier area at the beginning

of the study period. In our case the presented mass change estimates are compensated for that by using the temporal mean glacier area which includes glacier retreat (and also potential advance). Nevertheless, for the elevation change measurements itself it would be necessary to know the exact time of deglaciation during the study period of each raster cell to correct the elevation change rate accordingly.

On the other side it is difficult to quantify the actual influence of this effect on a regional scale as elevation changes are aggregated within elevation bands which might include areas which have become ice-free or have been ice-covered over the entire study period.

However the hypsometric distributions in Fig.2 show that in many subregions the most negative elevation changes are found at lowest glacier elevations which indicates that the surface lowering is “faster” than the glacier retreat.

→ In the case of the cited study, Fischer et al. 2015 compared a very long study period of ~30 years and starting in the 1980s which is probably one reason why their surface elevation change rates are less negative than ours.

→ Also added reference in L.204 (see last comment)

l.188-189: comparing your results to an estimate from the literature. How sensitive is this comparison to the value considered for the snow-ice density? Did you use the same conversion as the one used in the study you are comparing to?

→ Yes, most studies apply 850 kg/m³ (Huss 2013) as an approximate conversion factor for alpine type mountain glaciers. Only Paul & Haeberli (2008) used a higher value. Therefore we mentioned it in the text (l. 165).

l.193: ‘...is smaller’ à ‘are smaller’?

→ Corrected

l.198: explanation related to temperature change. See comment on l.168-169: glaciers also retreated during time period, which will also influence the mass balance, right?

→ Yes, glacier retreat during the observation period could be also an explanation. However we assume that in this case other effects are more important because the study periods of Zemp et al. (2019) and our study are rather similar (2006-2016 vs. 2000-2014). Therefore it seems to be more likely that the differences in the results are caused by the temperature change and other effects such as spatial coverage, uneven distribution of in-situ measurements and the extrapolation to the entire Alps and Pyrenees.

l.202-204: ‘strongest contributors are the ablation zones of large valley glaciers...’: indeed. May be worth mentioning that this is related to the fact that these are the glaciers that are most out of balance compared to the present-day climate due to their long response times: i.e. these lower parts of these large valley glaciers are relicts from older, colder time periods.

→ We added “...which are most out of balance and still adapting to present-day climate.”

Reviewer #2 (Remarks to the Author):

In the revised version of their manuscript, Sommer and colleagues have addressed all specific and detailed comments accordingly. They have changed the title to more focus on the observed changes. The authors have retained the future scenario analysis, although the importance of this estimation was reduced. With respect to processes like dynamic adjustment and climate change impacts on glacier mass balance, which are not considered in the approach, the results appear to be less robust compared to more detailed model studies. Conversely, the observed volume changes include processes of glacier vanishing such as basal melt and high mass losses due to the present state of low glacier dynamics in particular for many of the small to medium-size glaciers. Thus, the presentation of the results in Line 118ff, also stating the limitations, appears to be appropriate.

Thank you very much for the comments, we revised the uncertainty calculation of the elevation change measurements (see comments below).

Still, I'm concerned on the presented uncertainty of the surface elevation changes. Coregistration will eliminate a number of systematic errors of surface elevation changes (mean/median deviation, shifts). Nevertheless, the standard deviation of deviations between TandemX DEMs and plain surfaces was found to be in the order of several decimetres with RMSE in the order of meters (e.g. Becek et al. 2016, Wessel et al. 2018), and can be expected to be higher for SRTM DEMs (e.g. Kropáček et al. 2014). For their uncertainty estimation, the authors refer to the study of Rolstad et al. (2009). A fundamental parameter of this geostatistical analysis is the standard deviation of Δz ($\sigma\Delta z$). However, applying EQ. 2 using the presented lag distance of 312 m (L 407) and assuming a $\sigma\Delta z$ of 0.5 m for all glaciers of RGI6.0 Central Europe results in an average uncertainty of elevation changes of 0.4 m (i.e. 0.036 ma⁻¹), and 0.15 m (0.012 ma⁻¹) if weighted for glacier area. This is about 4 times to an order of magnitude higher compared to the uncertainties of surface elevation changes presented in Table 1. I agree that this simple calculation cannot match the results of the detailed analysis considering surface slopes. However, I recommend to present $\sigma\Delta z$ for a better reproducibility of uncertainty estimations in one or in a combination of the following ways:

→ We revised the uncertainties of the elevation changes and found an error within the decorrelation factor calculation which we unfortunately missed during the first revision. This bias influenced all regional uncertainties of results which were computed with two glacier inventories (beginning and end glacier area). We therefore calculated all uncertainties of the respective values again and replaced the values in the text and all figures/tables. The corrected uncertainties are in general higher, particularly in the smallest subregions where the majority of glaciers is small or very small.

→ Additionally, we followed the recommendations to change Fig.S8 and provide some further statistics on all non-glacier areas (see comments below).

(i) In Figure S8, the authors present the normalized median absolute deviation (NMAD), which is an error measure in particular for non-normal error distributions (Höhle and Höhle, 2009). If the distribution of the elevation differences between coregistered SRTM and TandemX DEM in stable areas is non-normal, this may be discussed. If the distribution is normal, $\sigma\Delta z$ can be presented instead of NMAD.

→ We replaced the NMAD error bars of Fig.S8 by the standard deviation per slope bin (filtered by 2-98% quantile) as used in the error analysis.

(ii) An additional Table (in the supplement) presenting the total area, total glacier

area, glacier area <35° slope, stable non-glacier areas <35° slope and $\sigma\Delta z$ of these areas for each of the 10 regions

→ We added an additional table in the supplement (Tab.S3) with statistics of non-glacier areas, including the glacier area ($\leq 50^\circ$ & $\leq 35^\circ$ slope), the non-glacier area ($\leq 50^\circ$ & $\leq 35^\circ$ slope) and the respective standard deviations (weighted by glacier area as used in the uncertainty analysis) of each subregion and each observation period.

(iii) An additional gridded data set with Δz of all stable non-glacier/coregistration areas considered for uncertainty estimation not masked to outlines of the RGI6.0 (compare Line 475)

→ We did not publish the entire raster datasets as the TanDEM-X CoSSc data is subject to regulations in terms of data use and the release of derived data products is limited to specific scientific applications (glacier change in our case).

If the uncertainties of surface elevation changes are found to be higher, uncertainties of (specific) mass change rates have to be adjusted accordingly.

Apart from this point, Sommer and colleagues present a well elaborated study of a first region-wide consistent analysis of glacier surface elevation changes over the European Alps worth to be published in Nature Communications. Figures and tables of the supplementary material complete the clear presentation of the results in the main text.

References

Becek, K., Koppe, W., Kutoğlu, Ş.H., 2016. Evaluation of Vertical Accuracy of the WorldDEM™ Using the Runway Method. Remote Sens., 8, 934. <https://doi.org/10.3390/rs8110934>

Höhle, J., Höhle, M., 2009. Accuracy assessment of digital elevation models by means of robust statistical methods. ISPRS J. Photogramm. Remote Sens. 64 (4),398-406, <https://doi.org/10.1016/j.isprsjprs.2009.02.003>

Kropáček, J., Neckel, N., Bauder, A., 2014: Estimation of Mass Balance of the Grosser Aletschgletscher, Swiss Alps, from ICESat Laser Altimetry Data and Digital Elevation Models. Remote Sens., 6, 5614-5632. <https://doi.org/10.3390/rs6065614>

Rolstad, C., Haug, T., Denby, B., 2009: Spatially integrated geodetic glacier mass balance and its uncertainty based on geostatistical analysis: application to the western Svartisen ice cap, Norway. J. Glaciol. 55, 666-680. <https://doi.org/10.3189/002214309789470950>

Wessel, B., Huber, M., Wohlfart, C., Marschalk, U., Kosmann, D., Roth, A., 2018: Accuracy assessment of the global TanDEM-X Digital Elevation Model with GPS data, ISPRS Journal of Photogrammetry and Remote Sensing, 139, 171-182. <https://doi.org/10.1016/j.isprsjprs.2018.02.017>

REVIEWERS' COMMENTS:

Reviewer #2 (Remarks to the Author):

In the recent version of their manuscript, Sommer and colleagues have revised the uncertainty calculation as suggested. They changed the uncertainty measure presented in Fig. S8 and added a table in the supplement showing the mean standard deviation between the coregistrated DEM. Thus, the presented uncertainties correspond to the applied methods and data used. My only minor comment is that in Tab. 1 almost all uncertainty values are updated except uncertainties in the column presenting the RGI6.0 based mass balance rate. These values should be revised, too. After revision of this single comment, I highly recommend this work for publication.

Reviewer #2 (Remarks to the Author):

In the recent version of their manuscript, Sommer and colleagues have revised the uncertainty calculation as suggested. They changed the uncertainty measure presented in Fig. S8 and added a table in the supplement showing the mean standard deviation between the coregistrated DEM. Thus, the presented uncertainties correspond to the applied methods and data used. My only minor comment is that in Tab. 1 almost all uncertainty values are updated except uncertainties in the column presenting the RGI6.0 based mass balance rate. These values should be revised, too. After revision of this single comment, I highly recommend this work for publication.

→ The error within the former uncertainty calculation of elevation change rates was related to the use of two glacier inventories (start & end area). However, for the column presenting RGI6.0 based results, the uncertainties were correct in the first place because we were using only one glacier area (RGI). For the same reason, the regional uncertainty values are a little bit lower because they only include the area determination error of one glacier inventory instead of two inventories.